# Redundant microtubule crosslinkers prevent meiotic spindle bending to ensure diploid offspring in *C. elegans*

Wenzhe Li[1], Helena A. Crellin[2], Dhanya Cheerambathur[2], Francis J. McNally[1]*

1 Department of Molecular and Cellular Biology, University of California, Davis, California, United States of America, 2 Wellcome Centre for Cell Biology & Institute of Cell Biology, School of Biological Sciences, University of Edinburgh, Edinburgh, United Kingdom

* fjmcnally@ucdavis.edu

**Data Availability Statement:** All data is included in the manuscript and its Supporting information files.

## Abstract

Oocyte meiotic spindles mediate the expulsion of ¾ of the genome into polar bodies to generate diploid zygotes in nearly all animal species. Failures in this process result in aneuploid or polyploid offspring that are typically inviable. Accurate meiotic chromosome segregation and polar body extrusion require the spindle to elongate while maintaining its structural integrity. Previous studies have implicated three hypothetical activities during this process, including microtubule crosslinking, microtubule sliding and microtubule polymerization. However, how these activities regulate spindle rigidity and elongation as well as the exact proteins involved in the activities remain unclear. We discovered that *C. elegans* meiotic anaphase spindle integrity is maintained through redundant microtubule crosslinking activities of the Kinesin-5 family motor BMK-1, the microtubule bundling protein SPD-1/PRC1, and the Kinesin-4 family motor, KLP-19. Using time-lapse imaging, we found that single depletion of KLP-19$^{KIF4A}$, SPD-1$^{PRC1}$ or BMK-1$^{Eg5}$ had minimal effects on anaphase B spindle elongation velocity. In contrast, double depletion of SPD-1$^{PRC1}$ and BMK-1$^{Eg5}$ or double depletion of KLP-19$^{KIF4A}$ and BMK-1$^{Eg5}$ resulted in spindles that elongated faster, bent in a myosin-dependent manner, and had a high rate of polar body extrusion errors. Bending spindles frequently extruded both sets of segregating chromosomes into two separate polar bodies. Normal anaphase B velocity was observed after double depletion of KLP-19$^{KIF4A}$ and SPD-1$^{PRC1}$. These results suggest that KLP-19$^{KIF4A}$ and SPD-1$^{PRC1}$ act in different pathways, each redundant with a separate BMK-1$^{Eg5}$ pathway in regulating meiotic spindle elongation. Depletion of ZYG-8, a doublecortin-related microtubule binding protein, led to slower anaphase B spindle elongation. We found that ZYG-8$^{DCLK1}$ acts by excluding SPD-1$^{PRC1}$ from the spindle. Thus, three mechanistically distinct microtubule regulation modules, two based on crosslinking, and one based on exclusion of crosslinkers, power the mechanism that drives spindle elongation and structural integrity during anaphase B of *C.elegans* female meiosis.

**Funding:** This work was supported by National Institute of General Medical Science grant R35GM136241 to FJM and Wellcome Trust Sir Henry Dale Fellowship (208833) to D.C. The funders had no role in study design, data collection and analysis, decision to publish, or preparation of the manuscript.

**Competing interests:** The authors have declared that no competing interests exist.

## Author summary

Meiosis reduces the number of chromosomes from four to one during the formation of egg and sperm, so that a fertilized egg has exactly two copies of each chromosome. Meiotic errors result in offspring with an incorrect number of chromosomes, which lead to prenatal death or birth defects. During mitosis, chromosome segregation is mediated by a bipolar mitotic spindle. Microtubules emanating from each spindle pole to the opposite ends of the cell generate pulling forces that must be resisted by the midzone microtubules of the spindle. In contrast, oocyte meiotic spindles are so small relative to the size of the oocyte that both spindle poles are subject to pulling forces from the same side of the oocyte. Normally, only one pole is pulled to the cortex to allow expulsion of half the chromosomes into a tiny cell called a polar body. Here we show that in the roundworm *C. elegans*, a specific subset of microtubule crosslinking proteins in the midzone are required to prevent the meiotic spindle from bending and expelling all chromosomes into polar bodies. This study reveals a new mechanism that ensures progeny's correct chromosome number.

## Introduction

Oocyte meiotic spindles mediate the expulsion of ¾ of the genome into polar bodies to generate diploid zygotes in nearly all animal species. Failures in this process result in aneuploid or polyploid offspring that are typically inviable. During meiosis, accurate meiotic chromosome segregation and polar body extrusion require the spindle to elongate while maintaining its structural integrity. Both mitotic and meiotic chromosome segregation in eukaryotes can be separated into mechanistically distinct phases, anaphase A and anaphase B. During anaphase A, chromosomes move toward the spindle poles; during anaphase B, the spindle elongates at the same velocity as chromosome separation [1–3]. During *C. elegans* female meiosis, the spindle shortens during anaphase A [4], and poleward chromosome movement is driven by kinetochore-dependent pulling forces that stretch the homologous chromosome pairs just before homolog separation, and the pulling forces are required for timely homolog separation [5]. During *C. elegans* female meiosis, anaphase B is driven by pushing forces from the microtubule array that elongates between separated homologs [6,7]. Anaphase B can push chromosomes apart in the apparent absence of metaphase spindle bipolarity [8] or sister chromatid cohesion [9]. Proteins required for meiotic chromosome separation have been identified [6,10] but the mechanism by which these proteins mediate anaphase remains unclear. In addition, electron tomography has revealed that *C. elegans* meiotic anaphase spindles are composed of extremely short microtubules [7,11]. This finding raises the question of how structural rigidity of the anaphase spindle is achieved to allow pushing between chromosomes.

In human RPE cells, anaphase is mediated by the redundant activities of the bipolar kinesin, Eg5, the KIF4A kinesin, and the microtubule crosslinker, PRC1 [12]. Despite their functions in human mitotic spindle elongation, their roles during meiotic anaphase in *C. elegans* are less clear. The *C. elegans* KIF4A homolog, KLP-19^KIF4A, has been shown to play a role in polar ejection force during prometaphase in mitosis [13] and meiosis [14]. Recently, KLP-19^KIF4A has been shown to act redundantly with kinetochore dynein to promote initial chromosome orientation [15]. However, the only post-metaphase defect reported was a morphologically altered telophase spindle [6]. The *C. elegans* PRC1 homolog, SPD-1^PRC1, has been reported to be responsible for pushing chromosomes apart during meiotic anaphase only in the absence of kinesin-14 motors [8]. Whereas KIF4A and PRC1 bind to each other and are interdependent

for localization on the mammalian mitotic midzone [16], it remains unknown whether their homologs, KLP-19[KIF4A] and SPD-1[PRC1], act as a complex during *C. elegans* meiotic anaphase.

Long-term depletion of Eg5 results in monopolar prometaphase spindles in mammalian mitosis [17–19] and meiosis [20]. Rapid small molecule inhibition of Eg5 in human RPE cells after metaphase has no effect on anaphase elongation rates but combining Eg5 inhibition with depletion of PRC1 or KIF4 dramatically slows down anaphase chromosome separation [12]. *C. elegans* carrying a deletion of the C-terminal half of the EG5 homolog, BMK-1, are viable and fertile [21]. In addition, in contrast with RPE cells, the *C. elegans bmk-1* deletion mutant exhibits faster than wild-type anaphase B in mitosis [22]. BMK-1 plays a redundant role in microtubule sorting during *C. elegans* meiotic spindle assembly and chromosome separation with cytoplasmic dynein and the kinesin-12 motor, KLP-18 [23].

In addition to chromosome segregation, another important feature of meiotic and mitotic spindles is their mechanical robustness in response to external forces. In human RPE1 cells, the combination of pulling force from cortical NuMA/dynein complex and pushing force from Kinesin Eg5 leads to a mechanically robust spindle resilient to external mechanical forces [24]. In *C. elegans* mitotic spindles, cortical dynein pulls on the astral microtubules emanating from the two spindle poles. In the absence of the PRC1 homolog, SPD-1[PRC1], or the central-spindlin component, CYK-4, cortical pulling forces rupture the central spindle [25]. Cortical pulling by dynein also ruptures mitotic spindles depleted of the kinetochore proteins HCP-1,2 or CLS-2 [26]. The maintenance of *C. elegans* meiotic spindle integrity in response to external forces has not been investigated previously. However, electron tomography [7,11] revealed that *C. elegans*, female meiotic spindles are composed of arrays of short microtubules with 40–50% of the spindle microtubules less than 1/4 of the spindle length. Such tiled arrays of microtubules likely require multiple microtubule crosslinkers to maintain structural integrity.

Other proteins shown to be required for *C. elegans* meiotic anaphase include ZYG-8[DCLK1] [4], MEL-28 [10], and CLS-2 [6,7]. The nucleoporin, MEL-28, promotes meiotic anaphase by recruiting protein phosphatase 1 [10] but the protein(s) that must be dephosphorylated by PP1 to drive anaphase have not been identified. ZYG-8[DCLK1] is a doublecortin kinase (DCLK1) homolog and DLCK1 has been shown to regulate motor binding to microtubules [27,28]. Its family members have also been shown to promote microtubule polymerization [29,30]. CLS-2 is a TOGL class microtubule-binding protein that might also drive anaphase by stimulating microtubule polymerization [6,7], however, loss of CLS-2 causes severe metaphase spindle defects, making interpretation of anaphase defects difficult [31].

We propose four protein-microtubule or protein-protein interaction activities that could regulate anaphase spindle integrity as well as drive elongation of microtubule bundles (Fig 1A). (1) Purified preparations of non-motor crosslinkers like PRC1 can stabilize microtubule bundles against externally applied forces. Because the protein friction generated by these cross-linkers are directionally asymmetric, thermal motion can drive contraction of these bundles [32–34]. (2) Purified preparations of microtubule motors like Eg5 can slide anti-parallel micro-tubules apart using ATP-driven motility [18]. (3) Purified microtubule-binding proteins can promote microtubule polymerization, which can exert a pushing force [35]. (4) Microtubule binding proteins can promote or inhibit binding of other proteins to microtubules through competition [27].

Here, we assayed anaphase B velocities and the spindle's resistance to bending in *C. elegans* meiotic embryos singly and doubly depleted of BMK-1[Eg5], SPD-1[PRC1], KLP-19[KIFA], and ZYG-8[DCLK1]. Our results indicate that BMK-1[Eg5], SPD-1[PRC1] and KLP-19[KIFA] regulate the meiotic spindle's mechanical robustness through partially redundant pathways, whereas ZYG-8[DCLK1] promotes anaphase B chromosome segregation through its regulations of SPD-1[PRC1]'s binding to microtubules.

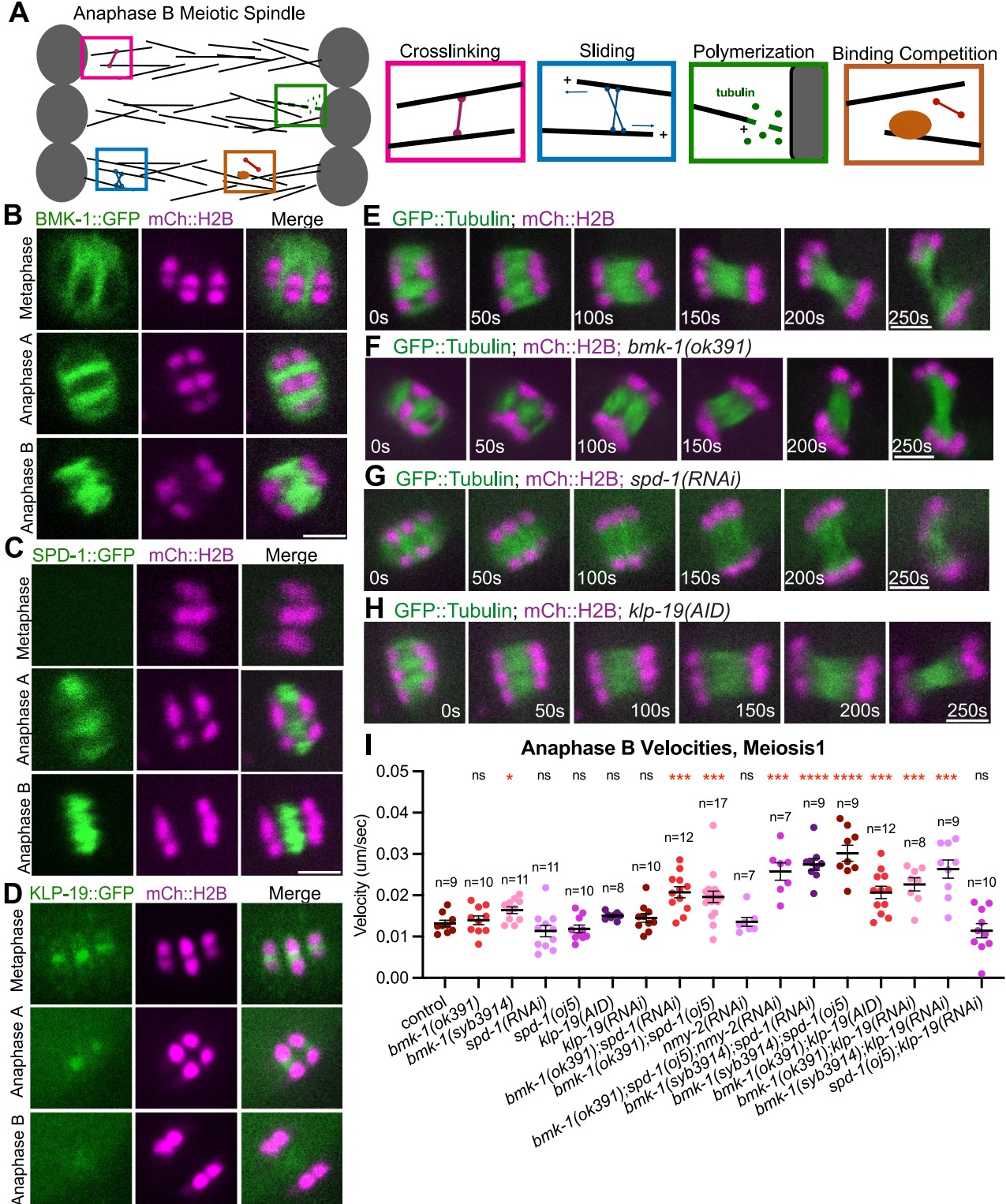

**Fig 1. BMK-1, SPD-1 and KLP-19 localize on the elongating anaphase meiotic spindle which is minimally affected by single depletions.** (A) Schematic of *C.elegans* oocyte meiotic anaphase B chromosome separation. Chromosomes (grey) and microtubules (black) are shown on the left. Four hypothetical activities that could regulate spindle integrity and/or elongation are shown on the right. (B-D) Representative images of the spindle region during metaphase, anaphase A and anaphase B, from time-lapse sequences of oocytes labeled with BMK-1::GFP and mCherry::H2B (B), SPD-1::GFP and mCherry::H2B (C), or KLP-19::GFP and mCherry::H2B (D). (E-H) Single-focal plane time-lapse images of control, *bmk-1(ok391), spd-1*

*(RNAi)* and *klp-19(AID)* with auxin embryos expressing GFP::Tubulin and mCherry::H2B. Time 0s for E-H and the rest of the paper is the time at anaphase B onset, where spindle reaches its shortest length and chromosomes reach the poles of the spindle. (I) Quantification of anaphase B chromosome separation velocity during meiosis 1 in control (strain with GFP::Tubulin and mCherry::H2B only) and mutant oocytes. Sample size (n), mean and standard error of the mean (SEM) are shown on the graph. Measurements were made in the period from 0–150 seconds after anaphase B onset. Statistics: Mann-Whitney U test (*p<0.05; ***p<0.001; ****p<0.0001). Statistical analyses were done between control and corresponding mutant oocytes. All scale bars = 3μm.

## Results

### BMK-1$^{Eg5}$, SPD-1$^{PRC1}$ and KLP-19$^{KIF4A}$ localize on the anaphase B spindle

To investigate the role of BMK-1$^{Eg5}$, SPD-1$^{PRC1}$ and KLP-19$^{KIF4A}$ in anaphase spindle elongation during *C. elegans* female meiosis, we first examined their localizations on the meiotic spindle using live imaging. For BMK-1$^{Eg5}$, we generated an in-frame GFP insertion at the endogenous *bmk-1* locus and revealed that BMK-1::GFP localized to spindle microtubules during metaphase (Fig 1B and S1A Fig). At anaphase A, BMK-1::GFP localized strongly at the midzone of the spindle but its intensity at the two poles of the shortening spindle was much lower relative to the mKate::tubulin signal (S1A Fig). BMK-1$^{Eg5}$ continued to localize on the spindle throughout anaphase B with no significant change in fluorescence intensity when measured as a ratio to background fluorescence (n = 11) (Fig 1B and S1B Fig).

To better understand the localization of SPD-1$^{PRC1}$, we examined a SPD-1::GFP transgenic strain [36] and found that SPD-1::GFP began to localize on the spindle midzone during anaphase A and anaphase B (Fig 1C). The width of the SPD-1::GFP fluorescence peak on the midzone was constant during anaphase B and was not affected by a BMK-1 deletion mutant (S1C and S1D Fig). Because the SPD-1$^{PRC1}$ homolog, PRC1, specifically crosslinks antiparallel microtubules in vitro [37,38], this result suggests that antiparallel microtubule overlap remains constant during spindle elongation.

In mammalian cells, kinesin-4's activity on the microtubule depends on its adaptor protein PRC-1. However, it remains unknown whether the *C. elegans* kinesin-4 homolog KLP-19$^{KIF4A}$ acts as a complex with the PRC-1 homolog SPD-1$^{PRC1}$. To investigate this, we first examined the localization of KLP-19::GFP tagged at the endogenous locus [39] on the spindle using live imaging. Consistent with previous reports, KLP-19::GFP initially localized at the mid-bivalent ring as well as the kinetochore cups during metaphase. As anaphase progressed, KLP-19::GFP dissociated from the mid-bivalent ring and localized weakly on the mid zone of the elongating spindle (Fig 1D). The localization of SPD-1$^{PRC1}$ was not altered in a BMK-1$^{Eg5}$ mutant or after KLP-19$^{KIF4A}$ depletion in the BMK-1$^{Eg5}$ mutant (S2A Fig). Likewise, the localization of KLP-19$^{KIF4A}$ was not altered in a BMK-1$^{Eg5}$ mutant or after SPD-1$^{PRC1}$ depletion in the BMK-1$^{Eg5}$ mutant (S2A Fig).

### Oocyte meiotic spindle elongation during anaphase is minimally affected by individual depletion of BMK-1$^{Eg5}$, SPD-1$^{PRC1}$ or KLP-19$^{KIF4A}$

To test for a requirement in anaphase B spindle elongation, we conducted time-lapse imaging of GFP::tubulin and mCherry::histone H2b during anaphase I and II in worms depleted of SPD-1$^{PRC1}$, KLP-19$^{KIF4A}$, or BMK-1$^{Eg5}$ (Fig 1E–1H). Anaphase B velocities were not significantly different than controls in *bmk-1(ok391)*, a deletion of the C-terminal half of the protein [40], although there was a slight but significant increase in anaphase B velocity in *bmk-1 (syb3914)*, a complete deletion of the *bmk-1* locus generated with CRISPR-Cas9 [23] (Fig 1I and S1E Fig). Neither *spd-1(RNAi)* nor *spd-1(oj5ts)* [41] worms had anaphase B velocities that were different than controls (Fig 1I; and S1E Fig). The efficiency of RNAi was confirmed by

the disappearance of SPD-1::GFP signal on the meiotic spindle with worms treated with *spd-1 (RNAi)* (S2C and S2D Fig). These results are consistent with previously published *spd-1(RNAi)* results on *C. elegans* meiotic spindles [6,7]. Anaphase B velocities were also not significantly different from controls when KLP-19$^{KIF4A}$ was depleted by RNAi or when KLP-19::AID::GFP [39] was depleted by auxin-induced degradation (Fig 1I and S1E Fig), consistent with a previous study that found no significant chromosome orientation, congression or segregation defects after KLP-19 depletion [15]. GFP fluorescence was not detected in KLP-19::AID::GFP worms after 20 hrs of auxin treatment, and the efficiency of RNAi was confirmed by the disappearance of KLP-19::GFP signal on the meiotic spindle with worms treated with *klp-19(RNAi)* (S2E and S2F Fig).

## Anaphase B spindle elongation is negatively regulated by redundant BMK-1$^{Eg5}$ and SPD-1$^{PRC1}$ activities

As individual depletion of BMK-1$^{Eg5}$, SPD-1$^{PRC1}$ or KLP-19$^{KIF4A}$ did not have strong effects on spindle elongation velocity, we hypothesized that two or more proteins may act in a functionally redundant manner. This possibility was suggested in a previous study in human RPE1 cells showing that mitotic spindle elongation during anaphase depended on the partially redundant activities of kinesin-5 and PRC-1 dependent kinesin-4 [12]. To test if similar mechanisms apply in *C.elegans* female meiosis, we performed double depletions of BMK-1$^{Eg5}$ using two different deletion alleles (*ok391* or *syb3914*) in combination with *spd-1(RNAi)* or *spd-1 (oj5)*. Interestingly, in double depletion of both BMK-1$^{Eg5}$ and SPD-1$^{PRC1}$, anaphase B spindles elongated faster than control spindles and faster than single depletion spindles (Figs 1I and 2A–2D and S1E Fig and S1–S3 Videos).

Whereas control and single depleted spindles elongated with a straight trajectory, double depletion of BMK-1$^{Eg5}$ and SPD-1$^{PRC1}$ resulted in a high frequency of spindles that either bent into a U shape or broke in the middle during anaphase B (Fig 2C and 2G, S3A, S3B and S3F Fig; S2 and S3 Videos). These defects were not an indirect consequence of spindle rotation defects as all anaphase spindles started perpendicular to the cortex before spindle bending. The apparent disruption of spindle integrity was accompanied by polar body extrusion defects. BMK-1$^{Eg5}$ SPD-1$^{PRC1}$ double depleted spindles exhibited a high frequency of failure to extrude chromosomes, extrusion of both sets of chromosomes into 2 polar bodies, or extrusion of unusually large polar bodies (Fig 2G and S3G Fig; S3 Video). Spindle bending in BMK-1$^{Eg5}$ and SPD-1$^{PRC1}$ double depleted spindles appeared to correlate with membrane invaginations which are myosin-dependent [42,43] and are associated with polar body formation. To test the hypothesis that BMK-1$^{Eg5}$ SPD-1$^{PRC1}$ double-depleted spindles are mechanically less rigid than controls and therefore more susceptible to bending forces derived from cortical myosin contraction, we examined BMK-1$^{Eg5}$ SPD-1$^{PRC1}$ NMY-2 triple depleted embryos. While Myosin (NMY-2) depletion had no effect on control spindle's elongation rate (Fig 1I), its depletion reduced the frequency of spindle bending in BMK-1$^{Eg5}$ SPD-1$^{PRC1}$ double depleted spindle (Fig 2E and 2F, and S3F Fig, p = 0.0034 meiosis I, p = 0.0068 meiosis II, Fisher's exact test) and resulted in a chromosome separation rate faster than the double mutant (Fig 1I and S1E Fig, p = 0.013 meiosis I, p = 0.033 meiosis II, Mann-Whitney U test; S4 Video). In addition, triple depleted spindles elongated in a straight trajectory but then broke with higher frequency than the double mutant (Fig 2F). Together, these results indicated that BMK-1$^{Eg5}$ and SPD-1$^{PRC1}$ act redundantly to slow anaphase B, and provide mechanical strength needed to resist forces applied by cortical myosin and prevent spindle bending, which can lead to extrusion of both or neither sets of chromosomes into polar bodies.

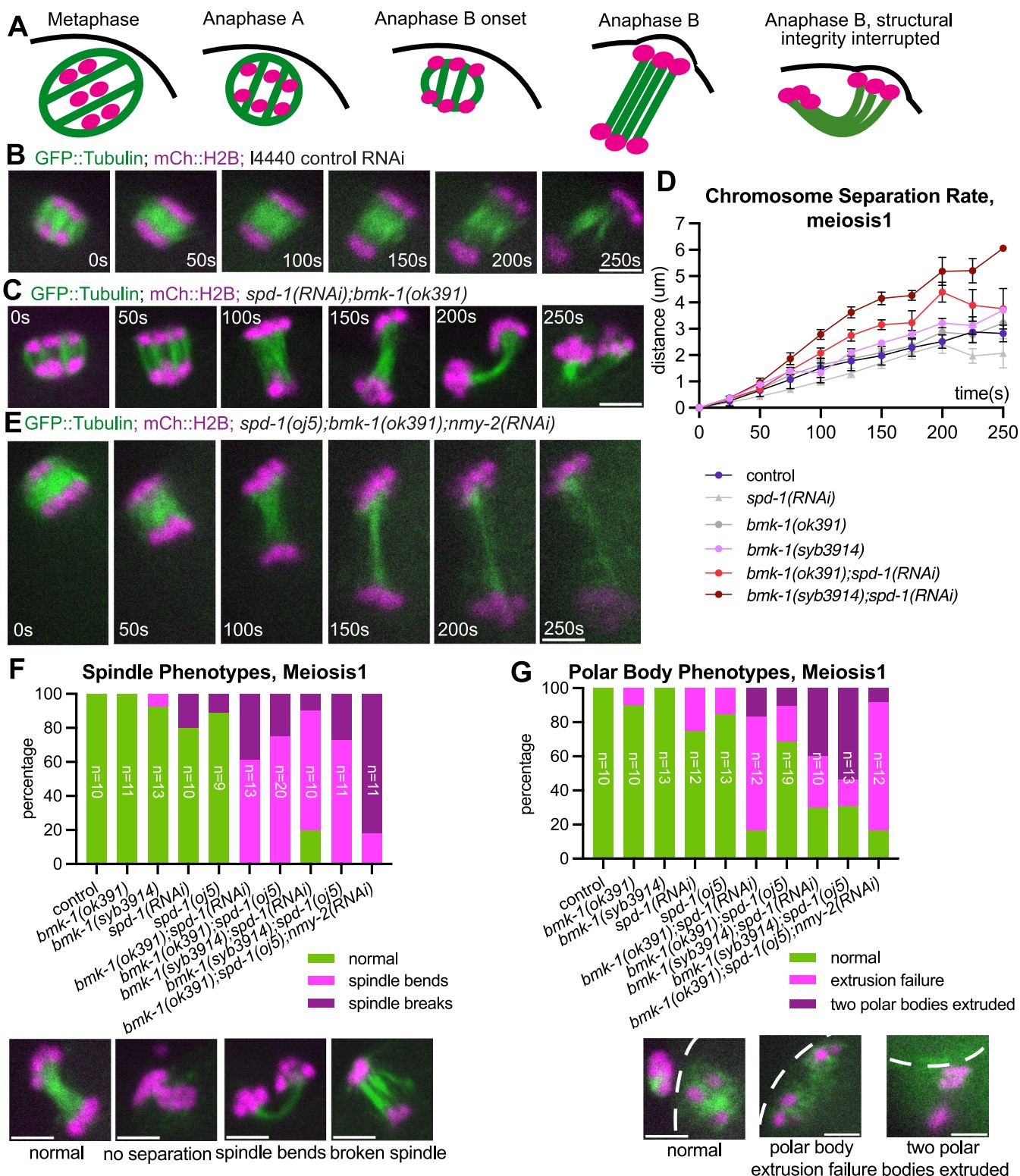

**Fig 2. Depletion of BMK-1 and SPD-1 results in faster anaphase B and mechanically fragile spindles.** (A) Schematic of *C.elegans* oocyte meiotic spindle structure in control and mutant oocytes. Chromosomes (magenta) and microtubules (green) are shown. (B-C) Single-focal plane time-lapse images of anaphase B control (B) and *spd-1(RNAi);bmk-1(ok391)* (C) meiotic spindles expressing GFP::Tubulin and mCherry::H2B. Time 0s = anaphase B onset. (D) Comparison of cumulative chromosome separation distance over time during meiosis 1 between control (n = 9), *spd-1(RNAi)* (n = 11), *bmk-1(ok391)* (n = 10), *bmk-1(syb3914)* (n = 11), *bmk-1(ok391);spd-1(RNAi)* (n = 13) and *bmk-1(syb3914);spd-1(RNAi)* (n = 9) during anaphase B, with average distance indicated and one standard deviation indicated. Time 0s = anaphase B onset. The 250s time point corresponds to roughly the end of anaphase in control oocytes. (E)

Single-focal plane time-lapse images of anaphase B *spd-1(oj5);bmk-1(ok391);nmy-2(RNAi)* spindle. Time 0s = anaphase B onset. (F, G) Top: percentage of spindle phenotypes (F) and polar body phenotypes (G) in control and mutant oocytes during meiosis 1. For both graphs, n for each condition is indicated on the bar of the condition. Bottom: representative images of spindle and polar body phenotypes. Images are from oocytes expressing GFP::Tubulin and mCherry::H2B. White dash lines indicate the locations of the cortex. All scale bars = 3μm.

## Anaphase B spindle elongation is negatively regulated by redundant BMK-1[Eg5] and KLP-19[KIF4A] activities

Previous studies suggested that human Kinesin 4 and its adaptor protein PRC1 acted as a complex in regulating spindle elongation [12,16]. To explore the possibility of *C. elegans* PRC1 homolog SPD-1[PRC1] acting as a complex with KLP-19[KIF4A] in regulating anaphase spindle integrity, we depleted KLP-19[KIF4A] using RNAi or KLP-19::AID::GFP with auxin-induced degradation, in combination with *bmk-1(ok391)* or *bmk-1(syb3914)*. Co-depletion of KLP-19[KIF4A] together with BMK-1[Eg5] mirrored the faster chromosome segregation and fragile spindle phenotype found after SPD-1[PRC1] and BMK-1[Eg5] double depletion during anaphase B (Fig 1I, S1E Fig, Fig 3A–3C; S4A, S4D and S4E Fig and S5 Video). This BMK-1[Eg5] KLP-19[KIF4A] double depletion resulted in a high frequency of spindle bending, spindle breakage, and abnormal polar body extrusion phenotypes (Fig 3D and S4F Fig). These defects were not an indirect consequence of spindle rotation defects as all anaphase spindles started perpendicular to the cortex before spindle bending. These phenotypes are similar to what we observed in BMK-1[Eg5] SPD-1[PRC1] double depletions, albeit milder in polar body extrusion phenotypes. These results indicated that BMK-1[Eg5] and KLP-19[KIF4A] limit meiotic spindle elongation through redundant pathways. On the other hand, double depletion of SPD-1[PRC1] and KLP-19 led to an anaphase B velocity that was not significantly different than control or single depletions (Fig 1I and S1E Fig).

It has also been reported that the direct interaction between PRC1 and CYK4, a subunit of the centralspindlin complex, is required for maintaining the mechanical robustness of microtubule bundles during cytokinesis in mitosis in human and in *C. elegans* [25,44]. We previously reported that single depletion of CYK-4 using RNAi resulted in no significant change in chromosome separation rate during anaphase [42]. To explore the possibility that *C.elegans* PRC1 homolog SPD-1[PRC1] works as a complex with CYK-4 in spindle regulation, we depleted CYK-4 using RNAi in combination with *bmk-1(ok391)*. We found no significant difference of anaphase B velocities between control and CYK-4 BMK-1[Eg5] double depleted spindles (S1F and S4B and S4C Fig) and a significant difference between *bmk-1(ok391); spd-1(RNAi)* vs *bmk-1(ok391); cyk-4(RNAi)* (p = 0.007 meiosis I, p = 0.001 meiosis II, Mann-Whitney U test). The efficiency of *cyk-4(RNAi)* was confirmed by the 100% polar body extrusion failure in the BMK-1[Eg5] CYK-4 double depleted embryos (S4F Fig), consistent with previously published *cyk-4(RNAi)* results on polar body extrusion [42]. These results suggested that CYK-4 is not in the same pathway as SPD-1[PRC1] in regulating anaphase B spindle integrity.

## The doublecortin family member ZYG-8[DCLK1] positively regulates anaphase B chromosome separation

Given that BMK-1[Eg5], KLP-19[KIF4A] and SPD-1[PRC1] all negatively regulate anaphase B spindle elongation, we next searched for proteins that promote anaphase B chromosome separation. We previously reported that the depletion of the doublecortin family member ZYG-8[DCLK1] reduced the chromosome separation rate during anaphase B [4]. In addition, ZYG-8[DCLK1] depletion was shown to reduce microtubule polymerization rate [30], a potential mechanism

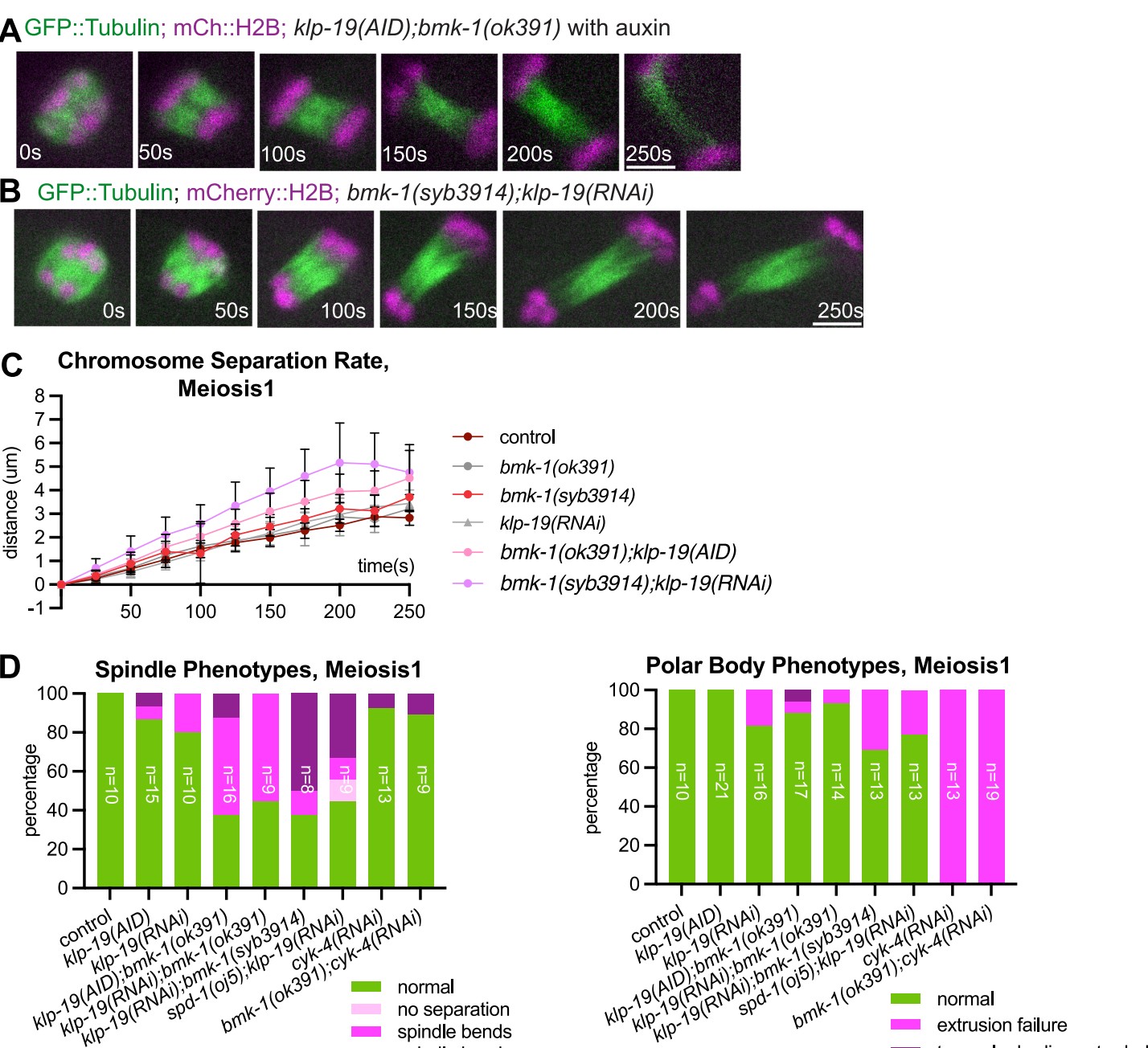

**Fig 3. Depletion of BMK-1 and KLP-19 results in faster anaphase B and mechanically fragile spindles.** (A-B) Single-focal plane time-lapse images of anaphase B *klp-19(AID);bmk-1(ok391)* with auxin (A) and *klp-19(RNAi); bmk-1(syb394)* (B) meiotic spindles expressing GFP::Tubulin and mCherry::H2B. Time 0s = anaphase B onset. (C) Comparison of cumulative chromosome separation distance over time during meiosis 1 between control (n = 9), *bmk-1(ok391)* (n = 10), *bmk-1(syb3914)* (n = 11), *klp-19(RNAi)* (n = 10), *bmk-1(ok391);klp-19(AID)* (n = 12) and *bmk*-1(syb3914);klp-19(RNAi) (n = 9) during anaphase B, with average distance indicated and one standard deviation indicated. Time 0s = anaphase B onset. The 250s time point corresponds to roughly the end of anaphase in control oocytes. (D) Percentage of spindle phenotypes and polar body phenotypes in control and mutant oocytes during meiosis 1. For both graphs, n for each condition is indicated on the bar of the condition. All scale bars = 3μm.

that drives meiotic spindle elongation during anaphase B. To further investigate ZYG-8$^{DCLK1}$'s role in anaphase B during meiosis, we examined the localization of GFP::ZYG-8 tagged at the endogenous locus. We found that GFP::ZYG-8 localized on the spindle microtubules during metaphase and anaphase A. During anaphase B, it localizes at the midzone of the elongating spindle (Fig 4A). To test ZYG-8$^{DCLK1}$'s role in anaphase B spindle elongation, we depleted the protein using 5 different methods. ZYG-8 RNAi depletion in a *zyg-8(ts)* background [*zyg-8 (RNAi);zyg-8(or484)*], GFP(RNAi) in a strain with endogenously tagged *gfp::zyg-8*, and auxin-induced degradation [*zyg-8(AID)*] resulted in 100% embryonic lethality whereas *zyg-8(RNAi)* alone or *zyg-8(or484)* alone resulted in less severe embryonic lethality phenotypes (S5A Fig). The 3 depletion methods with the strongest embryonic lethality resulted in a significant decrease in meiosis I anaphase B velocity (Fig 4B–4G; S6 and S7 Videos) as well as meiosis II anaphase B velocity (S5B and S5C Fig). We did not observe a significant decrease in anaphase B velocity when ZYG-8$^{DCLK1}$ was depleted with the two methods that resulted in only partial embryonic lethality, *zyg-8(RNAi)* only or *zyg-8(or484ts)* only at non-permissive temperature during meiosis I (Fig 4G) or meiosis II (S5B Fig). This is likely due to partial depletion or inactivation consistent with the hatch rates of the weaker alleles (S5A Fig).

As depletion of ZYG-8$^{DCLK1}$ had an opposite effect on spindle elongation compared to SPD-1$^{PRC1}$ and BMK-1$^{Eg5}$ or SPD-1$^{PRC1}$ and KLP-19$^{KIF4A}$ double depletions, we hypothesized that ZYG-8$^{DCLK1}$ functions at least partially by inhibiting the localization of BMK-1$^{Eg5}$, SPD-1$^{PRC1}$ or KLP-19$^{KIF4A}$ onto the spindle. To test this idea, we partially depleted ZYG-8$^{DCLK1}$ using RNAi in strains with BMK-1::GFP, SPD-1::GFP or KLP-19::GFP, respectively. We found that BMK-1::GFP signal intensity (S5D Fig) and KLP-19::GFP signal intensity (5A and 5B Fig) remained the same throughout both metaphase and anaphase when ZYG-8$^{DCLK1}$ is partially depleted. However, there was a significant increase of SPD-1::GFP signal intensity on the spindle throughout anaphase B of both meiosis I and II when ZYG-8$^{DCLK1}$ was partially depleted (Fig 5C and 5D). To test whether the observed effect was due to an increase in the amount of spindle microtubules, we measured the fluorescence intensity of GFP::tubulin in controls and worms depleted of ZYG-8. We found that there was no significant difference in the fluorescence intensity between control and *zyg-8(RNAi)* spindles or control and *zyg-8(or484)* spindles during anaphase B (S5E and S5F Fig). However, we found that spindles in embryos with *zyg-8(RNAi);zyg-8(or484)* have higher GFP::tubulin fluorescence intensity during anaphase B in meiosis I, and spindles in embryos with *zyg-8(AID)* have higher anaphase B GFP::tubulin fluorescence intensity in meiosis I and II. Because spindles in *zyg-8(RNAi)* embryos have increased intensity of SPD-1::GFP signal but not GFP::tubulin signal, we conclude that wild-type ZYG-8$^{DCLK1}$ may promote anaphase B spindle elongation in part by decreasing SPD-1$^{PRC1}$'s localization on the microtubules. The effect of the stronger depletions on GFP::tubulin intensity suggest that ZYG-8 also decreases the amount of microtubules on the meiotic spindle.

Because *zyg-8(RNAi)* has been reported to reduce microtubule polymerization rate [30], we hypothesized that *zyg-8(RNAi)* might slow down the accelerated anaphase B in BMK-1$^{Eg5}$ SPD-1$^{PRC1}$ double depleted embryos. Instead, in ZYG-8$^{DCLK1}$ BMK-1$^{Eg5}$ SPD-1$^{PRC1}$ triple depleted embryos we observed the same velocity increase as BMK-1$^{Eg5}$ SPD-1$^{PRC1}$ double depleted embryos (Fig 4F and 4G). We confirmed this result by depleting ZYG-8$^{DCLK1}$ using GFP RNAi on endogenously tagged GFP::ZYG-8 together with SPD-1$^{PRC1}$ and BMK-1$^{Eg5}$ depletion, and found that the resulting spindles elongate at a velocity not significantly different from the BMK-1$^{Eg5}$ SPD-1$^{PRC1}$ double depleted spindles (Fig 4G). This observation is consistent with our conclusion that ZYG-8$^{DCLK1}$ inhibits SPD-1$^{PRC1}$'s localization on the spindle to drive spindle elongation.

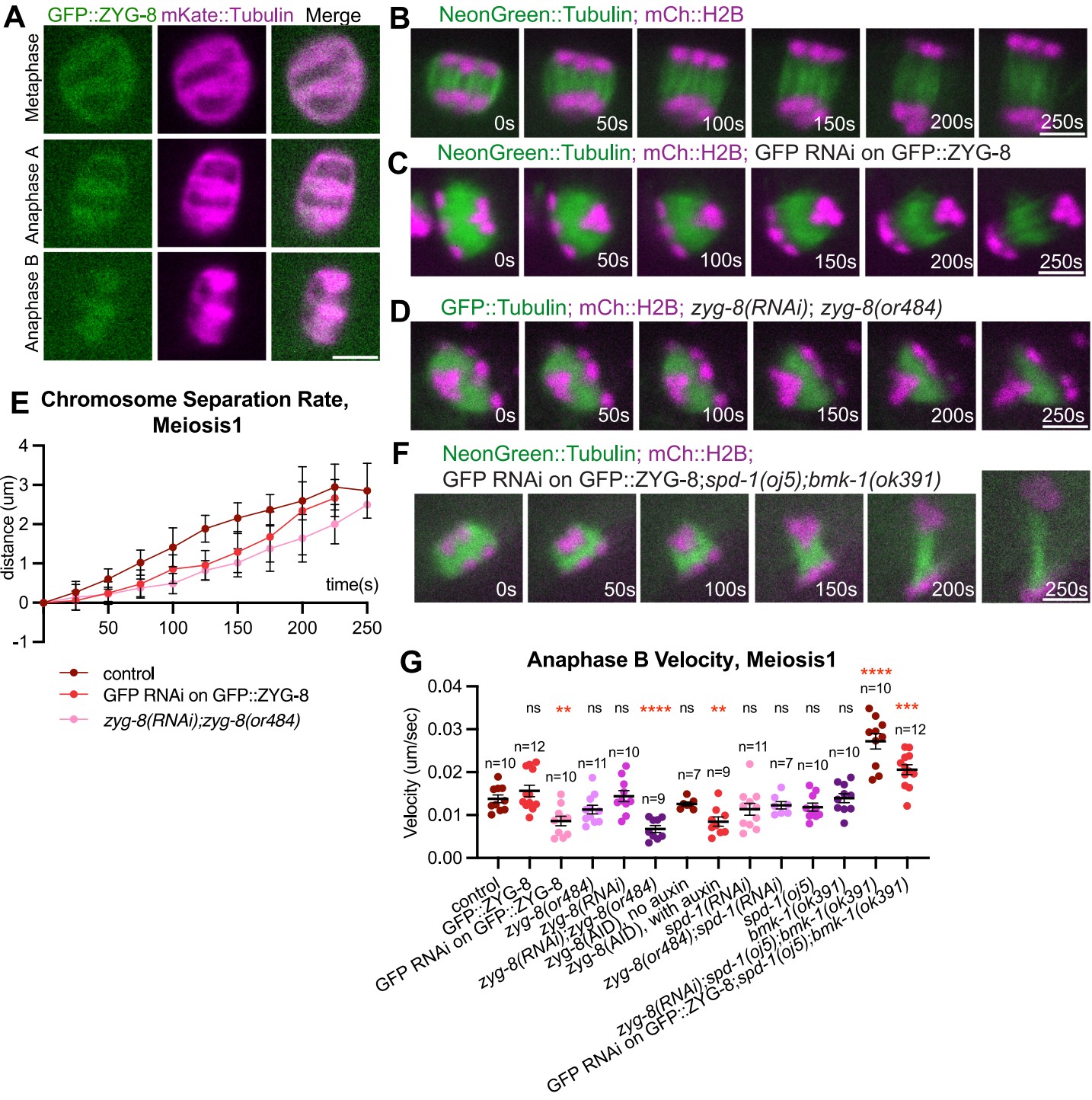

**Fig 4. ZYG-8 increases anaphase B velocity.** (A) Representative images of the meiotic spindle region during metaphase, anaphase A and anaphase B, from a time-lapse sequence of oocytes labeled with GFP::ZYG-8 and mKate::Tubulin. (B-D) Single-focal plane time-lapse images of control (B); GFP::RNAi on GFP::ZYG-8 (C); *zyg-8 (or484);zyg-8(RNAi)* (D) meiotic spindles. Time 0s = anaphase B onset. (E) Comparison of cumulative chromosome separation distance over time during meiosis 1 between control (strain with NeonGreen::Tubulin and mCherry::H2B only) (n = 10), GFP RNAi on GFP::ZYG-8 (n = 10) and *zyg-8(RNAi); zyg-8 (or484)* (n = 9) during anaphase B, with average distance indicated and one standard deviation indicated. Time 0s = anaphase B onset. The 250s time point corresponds to roughly the end of anaphase in control oocytes. (F) Single-focal plane time-lapse images of GFP RNAi on GFP::ZYG-8;*spd-1(oj5);bmk-1(ok391)* meiotic spindle expressing NeonGreen::Tubulin and mCherry::H2B. Time 0s = anaphase B onset. (G) Quantification of anaphase B chromosome separation velocity during meiosis 1 in control (strain with NeonGreen::Tubulin and mCherry::H2B only) and mutant oocytes. Sample size (n), mean and standard error of the mean (SEM) are shown on the graph. Measurements were made in the period from 0–150 seconds after anaphase B onset. Statistics: Mann-Whitney U test (**p<0.01; ****p<0.0001). Statistical analyses were done between control and corresponding mutant oocytes. All scale bars = 3μm.

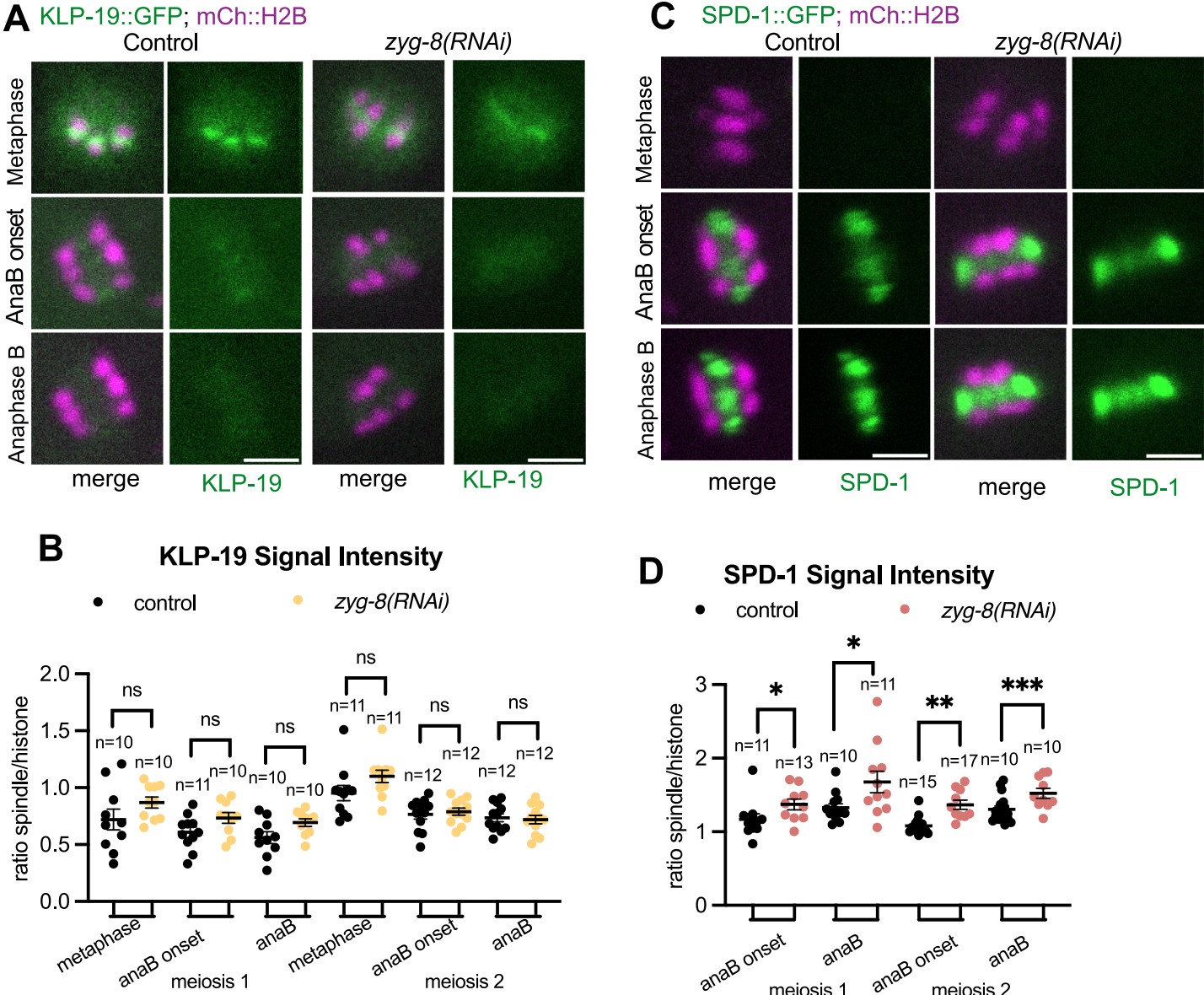

**Fig 5. ZYG-8 depletion leads to increased amount of SPD-1 on the spindle.** (A) Representative images of KLP-19::GFP's signal intensity on the spindle in control and *zyg-8(RNAi)* spindles. (B) Quantifications of KLP-19::GFP signal intensity on the spindle relative to mCherry::H2B intensity, at metaphase, anaphase B onset and anaphase B on control and *zyg-8(RNAi)* spindles. (C) Representative images of SPD-1::GFP's signal intensity on the spindle in control and *zyg-8(RNAi)* spindles. (D) Quantifications of SPD-1::GFP signal intensity on the spindle relative to mCherry::H2B intensity, at anaphase B onset and anaphase B on control and *zyg-8(RNAi)* spindles. Statistics: Mann-Whitney U test (*p<0.05; **p<0.01; ***p<0.001). All scale bars = 3μm.

## ZYG-8^DCLK1 stabilizes the metaphase spindle and prevents the anaphase A spindle from over-shortening

Metaphase spindle structure and anaphase A velocities appeared normal in single or double depletions of BMK-1^Eg5, SPD-1^PRC1, or KLP-19^KIF4A (Fig 7C and S6B Fig). However, it remains possible that subtle defects early in the cell cycle are responsible for the anaphase B phenotypes observed. In contrast, we observed a range of spindle morphology defects during both metaphase and anaphase A in ZYG-8^DCLK1 depleted embryos. During metaphase, *zyg-8*

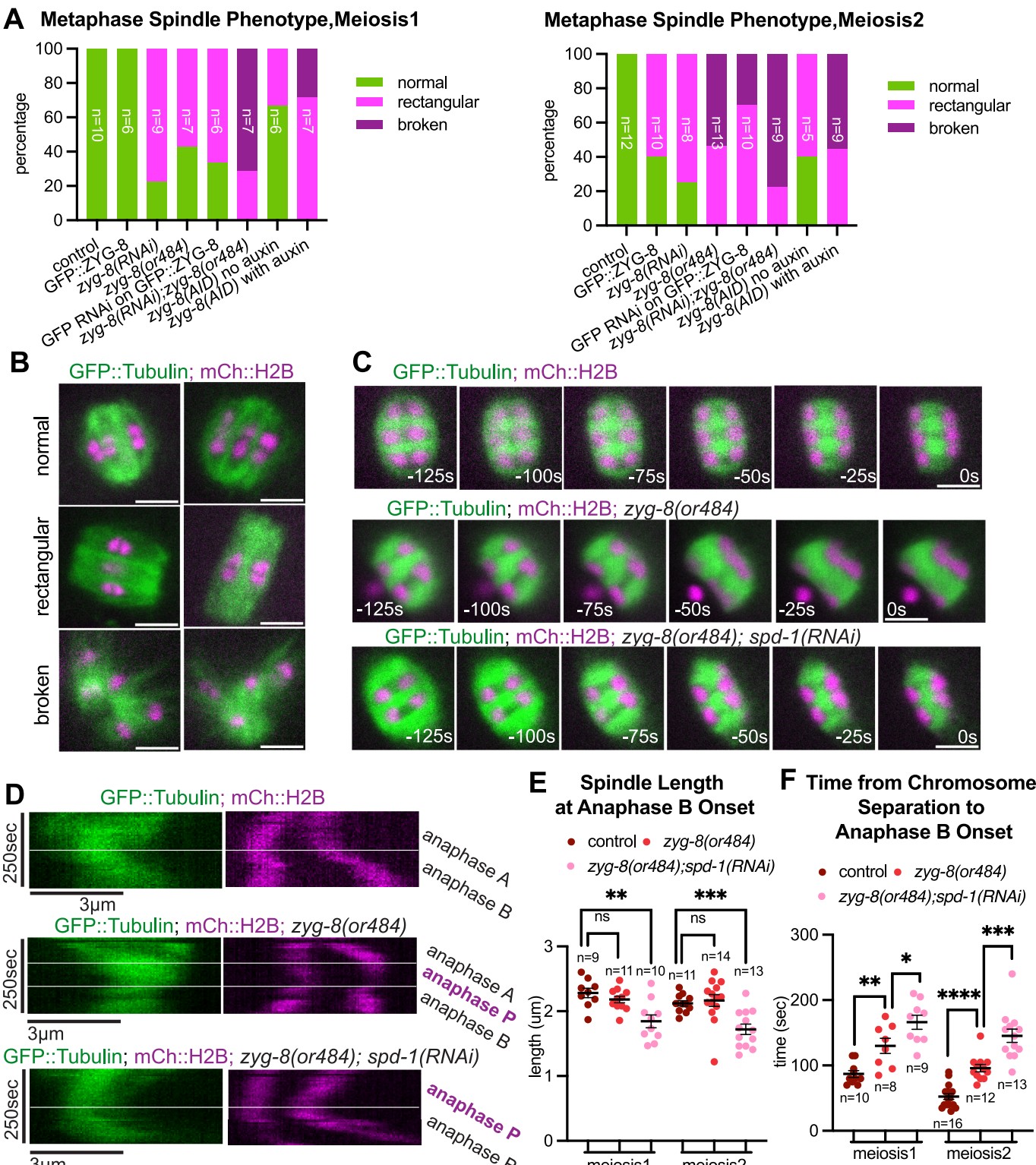

**Fig 6. ZYG-8 depletion leads to metaphase spindle morphology change and a chromosome segregation pause between anaphase A and anaphase B.** (A) Percentage of metaphase spindle phenotypes in control and mutant oocytes during meiosis 1 and 2. n for each condition is indicated on the bar of the condition. (B) Representative images of spindle phenotypes. Images are from oocytes expressing GFP::Tubulin and mCherry::H2B. (C) Single-focal plane time-lapse images of anaphase A control, *zyg-8(or484)* and *zyg-8(or484);spd-1(RNAi)* meiotic spindles expressing GFP::Tubulin and mCherry::H2B. Time 0s = anaphase B onset. The -125s time point corresponds to roughly the beginning of spindle rotation. (D) Kymograph analysis of chromosome separation in time-lapse sequences from control, *zyg-8(or484)* and *zyg-8(or484);spd-1(RNAi)* embryos. Start times are -125s to 125s relative to anaphase B onset. White dash lines indicate the beginning

of the phases labeled on the right. (E) Quantification of meiotic spindle length at anaphase B onset in control, *zyg-8(or484)* and *zyg-8(or484);spd-1(RNAi)* oocytes. Statistics: Mann-Whitney U test (**p<0.01; ***p<0.001). All scale bars = 3μm.

*(RNAi)*, *GFP(RNAi)* on endogenously tagged GFP::ZYG-8 worms, *zyg-8(AID)* on auxin, and *zyg-8(or484ts)* worms at nonpermissive temperature for 2 hours, all resulted in a rectangular metaphase spindle morphology as opposed to the oval shaped spindles observed in control (Fig 6A and 6B). These spindles were also unstable with a portion of spindles completely broke down during metaphase (Fig 6A and 6B). However, even the broken metaphase spindles were still able to form a bipolar anaphase B spindle and complete chromosome separation (S7 and S8 Videos). We also observed a fraction of the spindles with rectangular morphology in worms with endogenously tagged GFP::ZYG-8 or *zyg-8(AID)* with no auxin (Fig 6A and 6B). This is likely due to a partial loss of function of ZYG-8$^{DCLK1}$ due to the endogenous CRISPR tagging. Taken together, these results suggest that ZYG-8$^{DCLK1}$ plays a role in bipolar spindle stability during metaphase, but is dispensable for anaphase spindle bipolarity maintenance.

ZYG-8$^{DCLK1}$ depletion, either by *GFP(RNAi)* on endogenously tagged ZYG-8$^{DCLK1}$, *zyg-8 (AID)* on auxin, or *zyg-8(or484ts)* at non permissive temperature, resulted in a complex change in anaphase A kinetics compared to the controls (Fig 6C and 6D). We found that in ZYG-8$^{DCLK1}$ depleted embryos, chromosomes initially moved apart during anaphase A but stopped separating and plateaued in their distance for around 100 seconds before anaphase B onset (Fig 6D and S6, S7 and S9 Videos). This pause in between anaphase A and B is in contrast with the phenotype of ZYG-8$^{DCLK1}$ depletion during anaphase B, where the chromosome separation rate decreases uniformly throughout (Fig 4B–4E and S5C Fig). The plateaued stage in chromosome separation did not alter spindle length at anaphase B onset, but did result in an elongated time from spindle rotation completion to anaphase B onset (Fig 6E and 6F). We name this additional stage anaphase P, as chromosome separation pauses during this period between anaphase A and B (Figs 6C, 6D, 7A, 7B and S6A Fig). Because of the presence of anaphase P, the average velocities of chromosome separation 75 seconds before anaphase B onset are significantly lower than control during meiosis I and II except for *zyg-8(RNAi)* in meiosis I, which again may be due to partial protein depletion (Fig 7C and S6B Fig). These results suggest that ZYG-8$^{DCLK1}$ ensures that the anaphase A to B transition happens in a timely manner.

Because SPD-1$^{PRC1}$ localizes on the spindle as anaphase A progresses (Fig 1C), and SPD-1$^{PRC1}$'s human homolog PRC1 has been shown to have microtubule-bundling activities in vitro [45], a potential mechanism that regulates microtubule dynamics during the anaphase A to B transition, we hypothesized that SPD-1$^{PRC1}$ plays a role in regulating the anaphase A to B transition in addition to ZYG-8$^{DCLK1}$. We depleted both proteins on the spindle and found that the average time from chromosome separation to anaphase B onset was 166.1 seconds in meiosis I, 145.4 seconds in meiosis II in double depletions of *zyg-8(or484)* and *spd-1 (RNAi)* compared with 130.0 seconds in meiosis I, 95.8 seconds in meiosis II in ZYG-8$^{DCLK1}$ depletion alone (Fig 6F; S10 Video). In addition, chromosomes not only stopped separating but moved closer towards each other during anaphase P in the double depleted embryos (Figs 6C, 6D, 7A and 7B, S6A Fig). As a result, spindles were significantly shorter at anaphase B onset in ZYG-8$^{DCLK1}$ and SPD-1$^{PRC1}$ double depletion compared to the controls (Fig 6E). On the other hand, anaphase B velocity of ZYG-8$^{DCLK1}$ SPD-1$^{PRC1}$ double depleted spindles was not significantly different from the control (Fig 4G and S5B Fig). Taken together, these results suggest that ZYG-8 stabilizes spindle morphology during metaphase and anaphase A, and that SPD-

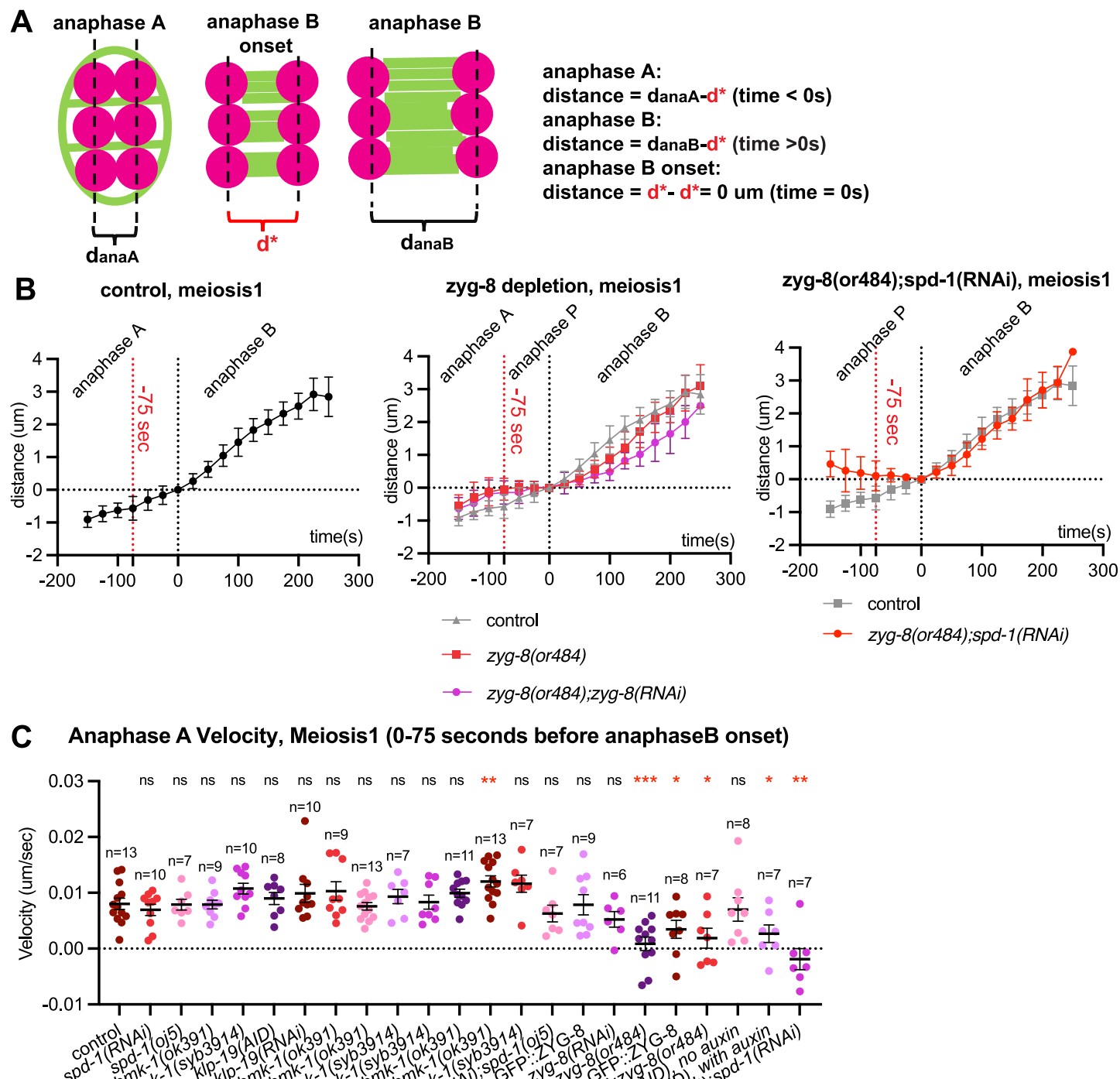

**Fig 7. Depletion of ZYG-8 and SPD-1 leads to chromosome movement reversal between anaphase A and anaphase B.** (A) Quantification of anaphase A chromosome separation velocity during meiosis 1 in control (strain with GFP::Tubulin and mCherry::H2B only) and mutant oocytes. Sample size (n), mean and standard error of the mean (SEM) are shown on the graph. Measurements were made in the period from 0–75 seconds before anaphase B onset. Statistics: Mann-Whitney U test (*p<0.05; ***p<0.001; ****p<0.0001). Statistical analyses were done between control and corresponding mutant oocytes. (B) Schematic of chromosome separation measurement between anaphase A and B. Chromosomes (magenta) and microtubules (green) are shown. (C) Comparison of cumulative chromosome separation distance over time during meiosis 1 in control (n = 19), *zyg-8(or484)* (n = 11), *zyg-8(or484);zyg-8(RNAi)* (n = 9) and *zyg-8(or484);spd-1(RNAi)* (n = 7) spindles from anaphase A to anaphase B using method described in (B), with average distance indicated and one standard deviation indicated. Time 0s = anaphase B onset. In the second and the third panels, the values for the control are also plotted in grey. Vertical dash lines indicate the locations of -75 seconds and 0 second on the plot, where velocity measurements in (A) where made.

1$^{PRC1}$ plays a role in anaphase A spindle dynamics in a pathway partially redundant with ZYG-8$^{DCLK1}$.

## Discussion

### Model for spindle integrity and elongation during anaphase B

Here we present a new model for *C. elegans* female meiotic spindle integrity and elongation (Figs 8 and 9). Integrity of the anaphase B spindle is maintained by BMK-1$^{Eg5}$, KLP-19$^{KIF4A}$, and SPD-1$^{PRC1}$ acting redundantly as microtubule crosslinkers. When crosslinking is reduced,

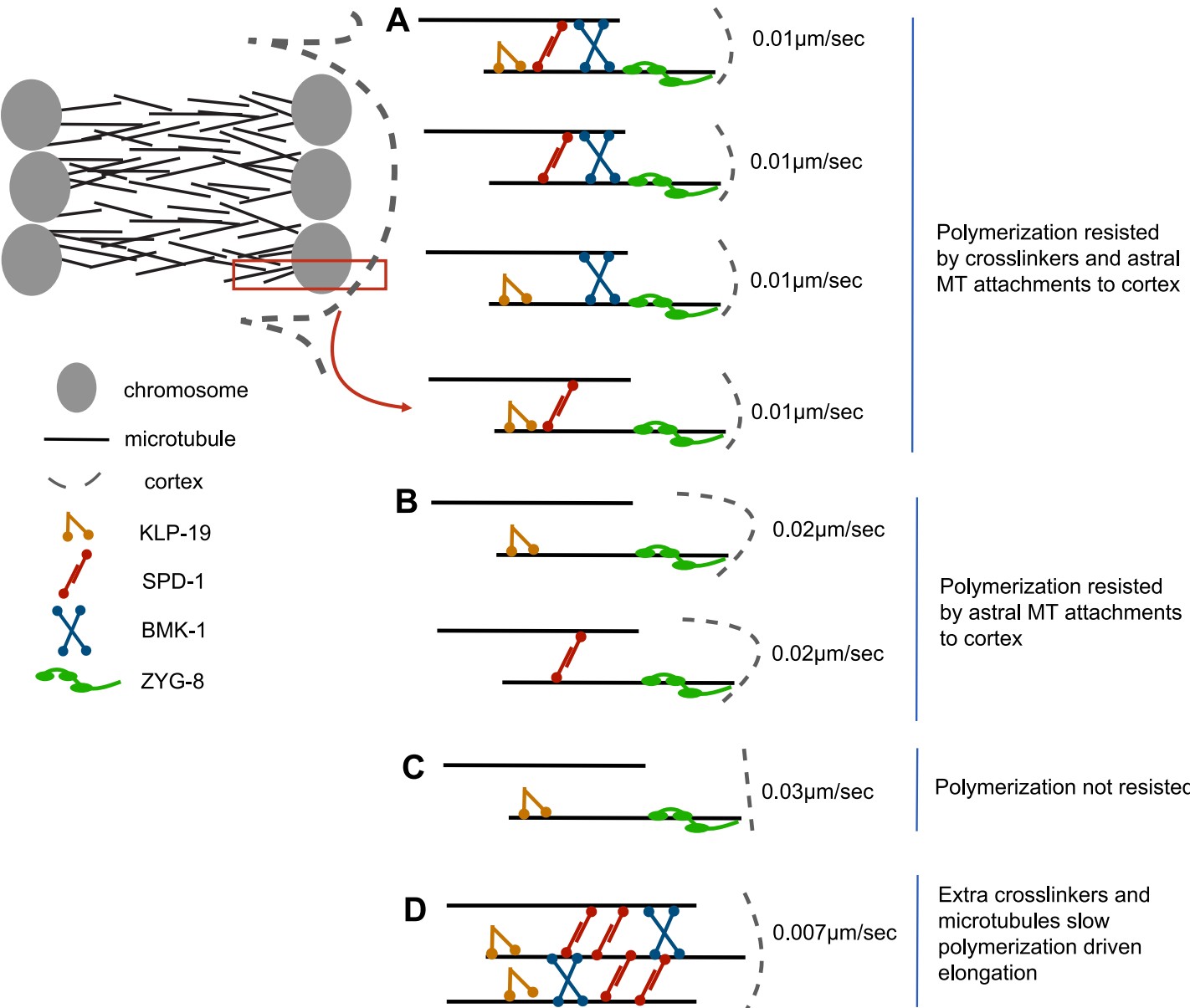

**Fig 8. Summary of anaphase B chromosome separation velocities.** (A) Spindle elongation resisted by crosslinkers and astral microtubules attaching to the cortex. (B) Spindle elongation resisted by astral microtubules attaching to the cortex only. (C) Spindle elongation not resisted; spindle elongates then breaks apart. (D) Spindle elongation resisted by extra crosslinkers.

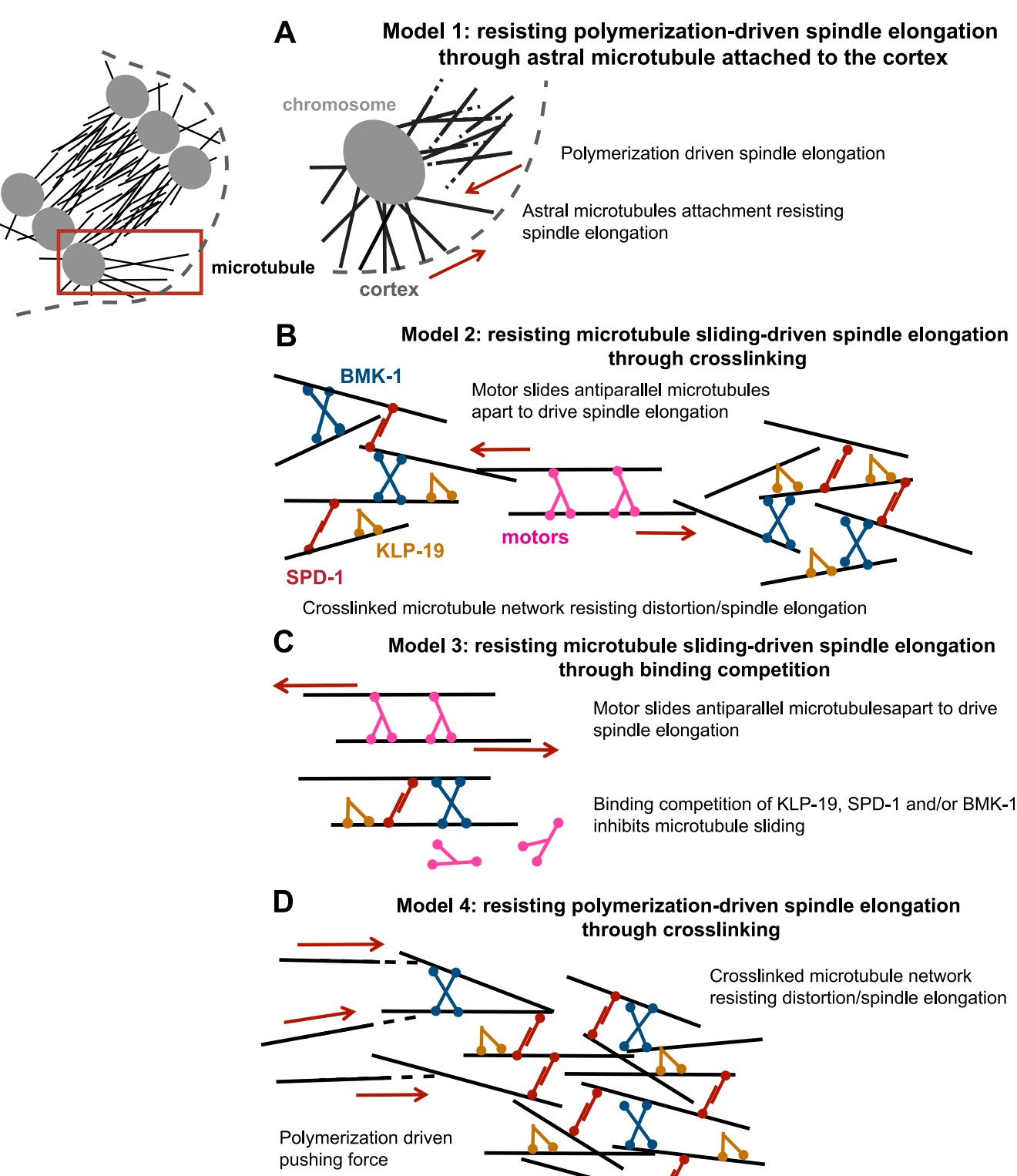

**Fig 9. Models of anaphase B chromosome separation regulation.** (A) Microtubule polymerization-driven spindle elongation resisted by inner pole astral microtubule attached to the proximal cortex. The region in the rectangle is magnified on the right. (B) Microtubule sliding-driven spindle elongation resisted by crosslinked microtubule networks. (C) Microtubule sliding-driven spindle elongation resisted by binding competition between molecular motors and crosslinkers. (D) microtubule polymerization-driven spindle elongation resisted by crosslinked microtubule networks.

the anaphase spindle is susceptible to myosin-dependent bending which can bring both sets of chromosomes close to the cortex. When both sets of chromosomes are in close proximity to the cortex, both sets of chromosomes are sometimes expelled in polar bodies, resulting in a haploid embryo. Spindle bending in double-depleted spindles occurs before myosin-dependent membrane invaginations are initiated [42] but after activation of dynein-dependent movement of one spindle pole to the cortex [46,47]. This suggests that spindle bending might be mediated by dynein-dependent pulling of both spindle poles toward the cortex and that myosin depletion reduces spindle bending by reducing cortical rigidity needed for cortical pulling. Directly testing the role of cortical pulling in spindle bending is difficult because depletion of cytoplasmic dynein affects spindle structure. In addition, the cortical-specific activators, GPR-1,2, are not required for recruiting dynein to the cortex during meiotic anaphase [46].

The need for microtubule crosslinkers to resist spindle deformation by external forces has been reported previously for mammalian mitosis [24]. In addition, double depletion of BMK-1$^{Eg5}$ and SPD-1$^{PRC1}$ or of BMK-1$^{Eg5}$ and KLP-19$^{KIF4A}$ resulted in faster anaphase B spindle elongation. These results suggest that microtubule crosslinking by BMK-1$^{Eg5}$, SPD-1$^{PRC1}$ and KLP-19$^{KIF4A}$ slows the rate of spindle elongation, and that spindle elongation is not driven by microtubule-microtubule sliding driven by the motor activities of BMK-1$^{Eg5}$ or KLP-19$^{KIF4A}$. The mechanism by which crosslinkers slow spindle elongation would depend on the mechanism of elongation. In *C. elegans* mitotic embryos, GPR-1,2-dependent cortical dynein pulls spindle poles in opposite directions through arrays of astral microtubules that reach opposite ends of the embryo. During mitosis, BMK-1 [22] and SPD-1$^{PRC1}$ [48] resist GPR-1,2-dependent cortical pulling. This situation has been reconstituted in vitro by using optical trapping to generate outward pulling on microtubule bundles. Microtubule crosslinking by purified preparations of the SPD-1$^{PRC1}$ homologs, Ase1 [34] or PRC1 [32], or the BMK-1 homolog, *Xenopus* kinesin-5 [33], resists these externally applied pulling forces in vitro. However, cortical pulling is unlikely to be a major contributor to anaphase B spindle elongation during normal meiosis because astral microtubules are unlikely to reach the distal cortex, and dynein depletion does not slow anaphase in meiosis [5,7]. However, it is still possible that astral microtubules from the inner pole reach towards the proximal cortex and resist spindle elongation (Fig 9A). It also remains possible that spindle elongation is driven by microtubule-microtubule sliding mediated by another kinesin. The kinesin 12 family member, KLP-18, is a strong candidate but analysis of anaphase after KLP-18 depletion has been hampered by the collapse of spindles into monopolar structures after KLP-18 depletion [14,49]. BMK-1$^{Eg5}$, KLP-19$^{KIF4A}$, and SPD-1$^{PRC1}$ might resist KLP-18-dependent microtubule sliding either by protein friction (Fig 9B) or by competitive binding (Fig 9C). Alternatively, anaphase B spindle elongation might be driven by CLS-2-dependent microtubule polymerization [6,7]. In vitro, the KLP-19$^{KIF4A}$ homolog, KIF4A, inhibits CLASP-driven microtubule polymerization, especially within microtubule bundles. This inhibition is driven by the fact that KIF4A/PRC1 has a higher binding affinity for bundled microtubules than CLASP [50]. KLP-19$^{KIF4A}$ might inhibit CLS-2-dependent anaphase B spindle elongation by a similar mechanism (Fig 9D).

The observation of similar phenotypes between SPD-1$^{PRC1}$ BMK-1$^{Eg5}$ double depleted spindles and KLP-19$^{KIF4A}$ BMK-1$^{Eg5}$ double depleted spindles leads to two plausible explanations. First, KLP-19$^{KIF4A}$ and SPD-1$^{PRC1}$ might work as a complex to crosslink antiparallel microtubules as has been suggested for their human homologs [16]. However, two results are not consistent with this idea during *C. elegans* meiosis. First, there is much more SPD-1$^{PRC1}$ than KLP-19$^{KIF4A}$ on the anaphase B spindle as it elongates and second, KLP-19$^{KIF4A}$ localization on the spindle is not dependent on SPD-1$^{PRC1}$. Our results instead suggest that SPD-1$^{PRC1}$ and KLP-19$^{KIF4A}$ work in separate pathways that both contribute to spindle integrity and slowing anaphase B. A recent study of anaphase B during *C. elegans* mitosis also suggested that SPD-

1[PRC1] and KLP-19[KIF4A] act in separate pathways. Depletion of KLP-19[KIF4A] prevented the GPR-1,2-dependent acceleration of anaphase B in SPD-1[PRC1] depleted embryos [51].

Removal of ZYG-8[DCLK1] caused an increase in the amount of SPD-1[PRC1] on the spindle and slowed anaphase B elongation, possibly due to increased protein friction generated by these crosslinkers. DCLK, the human homolog of ZYG-8[DCLK1], competes for motor binding in vitro [27]. However, unlike BMK-1[Eg5], SPD-1[PRC1], and KLP-19[KIF4A], ZYG-8[DCLK1] depletion also affected metaphase spindle shape and the transition from anaphase A to anaphase B, suggesting that its function on the spindle is not limited to binding competition during anaphase B. Depletion of ZYG-8[DCLK1] or double depletion of ZYG-8[DCLK1] and SPD-1[PRC1] prevented the smooth transition from anaphase A to anaphase B, by introducing an intermediate anaphase P (pause) during which inter-homolog distance was constant or actually reduced. The transition between anaphase A and anaphase B corresponds to the time that microtubule interactions with chromosomes switch from predominantly lateral to predominantly end-on with the inner face of separating univalents. The kinesin-13, KLP-7, is also required for this transition [52].

If ZYG-8[DCLK1] acts predominantly by competition for microtubule binding by other MAPs, KLP-18, CLS-2, and KLP-7 are potential candidates for competitors. In addition, ZYG-8[DCLK1] has a kinase domain in addition to its microtubule-binding domain, and its functions in the spindle may be to phosphorylate unknown proteins within the spindle. Analysis of a kinase dead ZYG-8[DCLK1] mutant should help resolve its mechanism of action.

## Materials and methods

### Generation of *C.elegans* strains

DKC40 was generated with CRISPR/Cas9-mediated genome editing at the endogenous *zyg-8* locus [53]. PHX5383 was generated with CRISPR/Cas9-mediated genome editing at the endogenous *bmk-1* locus by SunyBiotech. The remaining *C.elegans* strains were generated by standard genetic crosses of strains bearing previously published genome edits, and genotypes were confirmed by PCR and/or fluorescence microscopy. Genotypes of all strains are listed in S1 Table.

### Live *in utero* imaging

L4 larvae were incubated at 20˚C overnight on MYOB plates seeded with OP50. Worms were anesthetized by picking young adult hermaphrodites (for the best imaging quality) into a solution of 0.1% tricaine, 0.01% tetramisole in PBS in a watch glass for 30min as previously described [54,55]. Worms were then transferred in a small volume to a thin agarose pad (2% agarose in water) on a slide. Additional anesthetic in PBS solution was pipetted around the edges of the agarose pad, and a 22-x-30-mm coverslip was place on top. The slide was inverted and placed on the stage of an inverted microscope. Meiotic embryos were identified by brightfield microscopy before initiating time-lapse fluorescence. For all live imaging, the stage and immersion oil temperature was 21˚C-24˚C. For all time-lapse data, single-focal plane images were acquired with a Solamere spinning disk confocal microscope equipped with an Olympus IX-70 stand, Yokogawa CSU10, either Hamamatsu ORCA FLASH 4.0 CMOS (complementary metal oxide semiconductor) detector or Hamamatsu ORCA-Quest qCMOS (quantitative complementary metal oxide semiconductor) detector, Olympus 100x/1.35 objective, 100-mW Coherent Obis laser set at 30% power, and MicroManager software control. Pixel size was 65nm for the ORCA FLASH 4.0 CMOS detector and 46nm for the ORCA-Quest qCMOS detector. Exposures were 200ms for the ORCA FLASH 4.0 qCMOS detector and 100ms for the ORCA-Quest qCMOS detector. Time interval between image pairs was 5 seconds. Focus was

adjusted manually during time-lapse imaging. Only images in which both spindle poles were in focus were used for quantitative analysis. Both poles were considered to be in focus when both exhibited equal brightness and sharpness.

For documenting localizations of BMK-1::GFP, SPD-1::GFP, KLP-19::GFP and GFP::ZYG-8, time lapse images were taken at 5 second intervals on the entire meiotic process, and quantifications were done on the frames where the spindles were the most in focus at each stage.

## Timing

Control spindles maintain a steady-state length of 8 μm for 7min before initiating APC-dependent spindle shortening, followed by spindle rotation as spindle continues to shorten (anaphase A), and once spindle reaches its shortest length (anaphase B onset), it transitions to spindle elongation (anaphase B) [4,56]. Because both SPD-1 BMK-1 and KLP-19 BMK-1 double depletions only affect chromosome separation rate after anaphase B onset, we used time relative to anaphase B onset to compare parameters between control spindles and spindles with protein depletion(s). For temperature sensitive alleles, *C.elegans* were incubated at restrictive temperature (25°C) for 1.5 hours before filming at 22°C for up to three hours. The *zyg-8 (or484)* allele has been previously shown to be fast acting [57].

## Auxin induced degradation

*C.elegans* strains with *klp-19* endogenously tagged with auxin-inducible degrons and a TIR1 transgene were treated with auxin overnight on seeded plates. Auxin (indole acetic acid) was added to molten agar from a 400mM stock solution in ethanol to a final concentration of 4mM auxin before pouring plates, which were subsequently seeded with OP50 bacteria. Depletion of KLP-19::AID::GFP was confirmed by the disappearance of KLP-19::AID::GFP signals on the metaphase spindle shown in S1J Fig.

## RNAi

L3-L4 larvae were placed on IPTG-induced lawns of HT115 bacteria bearing L4440-based plasmids. For RNAi treatments of *spd-1 and klp-19*, worms were imaged live after 44–48 hours. For RNAi treatments of *nmy-2*, *cyk-4*, *zyg-8* and *GFP*, worms were imaged live after 24–26 hours. For RNAi treatments of *cls-2*, worms were imaged live both after 24–26 hours and after 44–48 hours, as indicated in text. RNAi clones for *spd-1*, *klp-19*, *nmy-2*, *cyk-4* and *cls-2* were from genomic RNAi feeding library (Medical Research Council Gene Services, Source BioScience, Nottingham, UK; [58]). *C.elegans* optimized GFP was cloned into L4440 for the *GFP(RNAi)* experiments.

## Anaphase B velocity measurements

All anaphase B velocity measurements were done on meiotic spindles filmed *in utero* at 5-second intervals. Anaphase B onset was defined as time 0 second when spindle reaches its shortest length. Distances between bivalent pairs or sister chromatids on each spindle were then measured by drawing a line connecting the center of the pair of the most in focus chromosomes and measuring the length of the line in pixel value. In cases where the spindle elongated in a straight line, a straight line was drawn along the spindle to connect the two chromosomes; in cases where spindle bent and the fluorescently tagged microtubules in between were clearly visible, a segmented line was drawn tracing the bending spindle to connect the two chromosomes. The pixel values were then converted to μm in length depending on the camera used for live imaging. Measurements were made every 25 seconds (5 frames) for 200–250 seconds,

until meiosis II prometaphase onset (for meiosis I) or pronuclei formation (for meiosis II). For each meiotic spindle, measurements in distance at each time point were then subtracted by the measurement in distance at time 0 second as an indication of separation since anaphase B onset, before they were grouped together for row statistics analysis (Figs 2D, 3C and 4C). Anaphase B velocities for each spindle were calculated by first subtracting distance at time 0 second from distance at time 150 seconds, indicating the total movement of chromosomes during time 0–150 seconds. The distance was then divided by 150 for the resulting velocities in μm per second. Time point at 150 seconds was chosen because at this time point, chromosomes have moved away from each other for a relatively long distance but tend to stay on the same focal plane as the one at time 0 second.

### Anaphase A velocity measurements

All anaphase A velocity measurements were done on the same meiotic spindle movies where anaphase B velocity measurements were made, and measurements in S4F Fig included only chromosomes separating parallel with the focal plane. Spindles with end-on orientations during anaphase A were not included for measurements. Measurements were made every 25 seconds (5 frames) for 100–150 seconds before anaphase B onset (defined as time 0 second), after the spindle started shortening and chromosomes began moving apart. Distance between bivalent pairs or sister chromatids on each spindle were then measured by drawing a line connecting the center of the pair of the most in focus chromosomes and measuring the length of the line in pixel value. The pixel values were then converted to μm in length depending on the camera used for live imagining.

### Fluorescence intensity measurements

Fluorescence intensity measurements in Figs S1B, S1D, S1E, S1F, S1H and S1J, 4F and 4H, S4C and S4E, are from single-focal plane images chosen from time-lapse imaging movies. Quantification of fluorescence intensity change in BMK-1::GFP, SPD-1::GFP and KLP-19::GFP at different stages were done on chosen single focal plane images in corresponding stages where the spindle is the most in focus. A spindle was judged to be the most in focus with the highest pixel intensity and sharpest edges. BMK-1::GFP, SPD-1::GFP, KLP-19::GFP and GFP::Tubulin intensities on the spindle in Figs S1B, S1E, S1F and S1H, 4F and 4H, S4C and S4E were measured using the Polygon Tool (ImageJ software) to outline the entire spindle. For each spindle measurement, the mean mCherry::histone H2b signal of the most in-focus chromosome pair in the same frame was also measured. KLP-19::GFP intensity on the chromosome during metaphase in S1J Fig was measured by tracing the most in-focus chromosome pair on the metaphase spindle using the Polygon Tool to obtain the mean pixel value of the area. For each chromosome pair, the mCherry::histone H2B signal was also measured. For both cases, mean pixel values of the spindle or chromosome pairs and histone signal were determined, and the value on the spindle or chromosome pairs were then divided by the value of the histone signal to generate a ratio for comparison. This ratio should remove any inconsistencies with spindle depth and the depth of the oocyte within the *C.elegans* uterus.

### Statistics

P values were calculated in GraphPad Prism using the Mann-Whitney U test for comparing means of only two groups, the Kruskal-Wallis test comparing means of three or more groups, and the Fisher's exact test analyzing contingency tables of spindle and polar body phenotypes. Correction for multiple comparisons to one control was used, rather than correction for all possible comparisons.

## Supporting information

**S1 Fig. BMK-1, SPD-1 and KLP-19 localize on the meiotic spindle at distinct stages.** (A) Representative images of oocyte meiotic spindle region during metaphase, anaphase A and anaphase B, from a time-lapse sequence of BMK-1::GFP and mKate::Tubulin. (B) BMK-1:: GFP signal intensities on the entire spindle were determined relative to mCherry::H2B intensities. (C) Representative images of SPD-1::GFP's signal width on the spindle throughout anaphase B. (D) top: schematic of SPD-1::GFP signal width determination on anaphase B spindles. Chromosomes (magenta) and SPD-1::GFP (green) are shown. Bottom: quantifications of SPD-1::GFP signal width at different timepoints throughout anaphase B of meiosis 1 and meiosis 2 on control and *bmk-1(ok391)* spindles. (E) Quantifications of anaphase B chromosome separation velocity during meiosis 2 in control (strain with GFP::Tubulin and mCherry::H2B only) and mutant oocytes. Sample size (n), mean and standard error of the mean (SEM) are shown on the graph. Statistics: Mann-Whitney U test (*p<0.05; **p<0.01; ***p<0.001; ****p<0.0001). Statistical analyses were done between control and corresponding mutant oocytes. (F) Quantifications of anaphase B chromosome separation velocity during meiosis 1 and 2 in control, *cyk-4 (RNAi)* and *bmk-1(ok391); cyk-4(RNAi)*. Sample size (n), mean and standard error of the mean (SEM) are shown on the graph. All scale bars = 3µm.
(EPS)

**S2 Fig. SPD-1 and KLP-19 localize on the meiotic spindle independently of BMK-1 and each other.** (A) Quantifications of SPD-1::GFP signal intensity on the spindle relative to mCherry::H2B intensity, at anaphase B onset and anaphase B on control, *bmk-1(ok391)* and *klp-19(RNAi);bmk-1(ok391)* spindles. (B) Quantifications of KLP-19::GFP signal intensity on the spindle relative to mCherry::H2B intensity, at anaphase B onset and anaphase B on control, *bmk-1(ok391)* and *spd-1(RNAi);bmk-1(ok391)* spindles. (C) Representative images of SPD-1:: GFP's signal intensity on the spindle in control and *spd-1(RNAi)* spindles. (D) Quantification of SPD-1::GFP 's signal intensity relative to cytoplasm during anaphase B on the control (strain with SPD-1::GFP, mCherry::H2B) and *spd-1(RNAi)* spindles. (E) Representative images of KLP-19::GFP's signal intensity on the spindle in control and *KLP-19(RNAi)* spindles. (F) Quantification of KLP-19::GFP 's signal intensity on the chromosome pairs relative to cytoplasm during metaphase on the control (strain with KLP-19::GFP; mCherry::H2B; mKate:: Tubulin), *klp-19(RNAi)* and *klp-19(AID)* with auxin spindles. All scale bars = 3µm.
(EPS)

**S3 Fig. Depletion of BMK-1 and SPD-1 results in faster anaphase B and mechanically fragile spindles.** (A-B) Single-focal plane time-lapse images of anaphase B *spd-1(oj5);bmk-1 (ok391)* (A) and *spd-1(RNAi);bmk-1(syb3914)* (B) meiotic spindles expressing GFP::Tubulin and mCherry::H2B. Time 0s = anaphase B onset. (C) Comparison of cumulative chromosome separation distance over time during meiosis 2 between control (n = 11), *bmk-1(syb3914)* (n = 9), *bmk-1(ok391);spd-1(RNAi)* (n = 9) and *bmk-1(syb3914);spd-1(RNAi)* (n = 9) during anaphase B, with average distance indicated and one standard deviation indicated. (D) Comparison of cumulative chromosome separation distance over time during meiosis 1 and 2 between control (meio1 n = 9; meio2 n = 11), *spd-1(RNAi)* (meio1 n = 11; meio2 n = 8), *bmk-1 (ok391)* (meio1 n = 10; meio2 n = 9) during anaphase B, with average distance indicated and one standard deviation indicated. (E) Percentage of spindle phenotypes and polar body phenotypes in control and mutant oocytes during meiosis 2. For both graphs, n for each condition is indicated on the bar of the condition. All scale bars = 3µm.
(EPS)

**S4 Fig. Depletion of BMK-1 and KLP-19 results in faster anaphase B and mechanically fragile spindles.** (A-C) Single-focal plane time-lapse images of anaphase B *klp-19(RNAi);bmk-1(ok391)* (A), *cyk-4(RNAi)* (B) and *cyk-4(RNAi);bmk-1(ok391)* (C) meiotic spindles expressing GFP::Tubulin and mCherry::H2B. Time 0s = anaphase B onset. (D) Comparison of cumulative chromosome separation distance over time during meiosis 2 between control (n = 11), *bmk-1 (syb3914)* (n = 9), *bmk-1(ok391);klp-19(AID)* (n = 14) and *bmk-1(syb3914);klp-19(RNAi)* (n = 7) during anaphase B, with average distance indicated and one standard deviation indicated. (E) Comparison of cumulative chromosome separation distance over time during meiosis 1 and 2 between control (meio1 n = 9; meio2 n = 11), *bmk-1(ok391)* (meio1 n = 10; meio2 n = 9) and *klp-19(RNAi)* (meio1 n = 10; meio2 n = 9) during anaphase B, with average distance indicated and one standard deviation indicated. (F) Percentage of spindle phenotypes and polar body phenotypes in control and mutant oocytes during meiosis 2. For both graphs, n for each condition is indicated on the bar of the condition. All scale bars = 3μm.
(EPS)

**S5 Fig. ZYG-8's activity during anaphase B is not due to an increased tubulin amount on the spindle.** (A) Summary of embryonic viability in broods produced by hermaphrodites of indicated genotypes, cultured at 20˚C unless indicated otherwise. (B) Quantification of anaphase B chromosome separation velocity during meiosis 2 in control (strain with NeonGreen::Tubulin and mCherry::H2B only) and mutant oocytes. Sample size (n), mean and standard error of the mean (SEM) are shown on the graph. Statistics: Mann-Whitney U test (**p<0.01; ****p<0.0001). Statistical analyses were done between control and corresponding mutant oocytes. (C) Comparison of cumulative chromosome separation distance over time during meiosis 2 between control (strain with NeonGreen::Tubulin and mCherry::H2B only) (n = 11), GFP RNAi on GFP::ZYG-8 (n = 12) and *zyg-8(RNAi); zyg-8 (or484)* (n = 11) during anaphase B, with average distance indicated and one standard deviation indicated. Time 0s = anaphase B onset. (D) Quantifications of BMK-1::GFP signal intensity on the spindle relative to mCherry::H2B intensity, at anaphase B onset and anaphase B in control and mutant spindles. (E) Representative images of the meiotic spindle region during metaphase, anaphase A and anaphase B of control and mutant oocytes, expressing GFP::Tubulin and mCherry::H2B (not shown). (F) Quantification of GFP::Tubulin intensities on the meiotic spindle relative to mCherry::H2B in control and *zyg-8(RNAi)* oocytes at anaphase B onset and anaphase B. All scale bars = 3μm.
(EPS)

**S6 Fig. Depletion of ZYG-8 and SPD-1 leads to chromosome movement reversal between anaphase A and anaphase B.** (A) Quantification of anaphase A chromosome separation velocity during meiosis 2 in control and mutant oocytes. Sample size (n), mean and standard error of the mean (SEM) are shown on the graph. Measurements were made in the period from 0–75 seconds before anaphase B onset. Statistics: Mann-Whitney U test (*p<0.05; **p<0.01; ***p<0.001; ****p<0.0001). Statistical analyses were done between control and corresponding mutant oocytes. (B) Comparison of cumulative chromosome separation distance over time during meiosis 2 in control (n = 22), *zyg-8(or484)* (n = 14), *zyg-8(or484);zyg-8 (RNAi)* (n = 11) and *zyg-8(or484);spd-1(RNAi)* (n = 8) spindles from anaphase A to anaphase, with average distance indicated and one standard deviation indicated. Time 0s = anaphase B onset. In the second and the third panels, the values for the control are also plotted in grey. Vertical dash lines indicate the locations of -75 seconds and 0 second on the plot, where velocity measurements in (A) were made.
(EPS)

**S1 Table. *C.elegans* Strain List.** List of genotypes of all strains used in this paper.
(DOCX)

**S1 Data. Numerical values for all graphs shown in this paper.**
(XLSX)

**S1 Video. Anaphase B chromosome separation and spindle elongation in control oocytes.**
*In utero* time-lapse movie with GFP::Tubulin and mCherry::H2B labeled.
(AVI)

**S2 Video. Anaphase B chromosome separation and spindle elongation in a strain with *bmk-1(ok391); spd-1(RNAi).*** *In utero* time-lapse movie with GFP::Tubulin and mCherry:: H2B labeled.
(AVI)

**S3 Video. Anaphase B chromosome separation and spindle elongation in a strain with *bmk-1(ok391); spd-1(RNAi).*** *In utero* time-lapse movie with GFP::Tubulin and mCherry:: H2B labeled. Both chromosome populations were extruded as polar bodies.
(AVI)

**S4 Video. Anaphase B chromosome separation and spindle elongation in a strain with *bmk-1(ok391); spd-1(oj5); nmy-2 (RNAi)* at 25˚C.** *In utero* time-lapse movie with GFP:: Tubulin and mCherry::H2B labeled.
(AVI)

**S5 Video. Anaphase B chromosome separation and spindle elongation in a strain with *bmk-1(ok391); klp-19(AID)* with auxin.** *In utero* time-lapse movie with GFP::Tubulin and mCherry::H2B labeled.
(AVI)

**S6 Video. Anaphase A and B chromosome separation and spindle morphology change in a strain with GFP RNAi on endogenously tagged GFP::ZYG-8.** *In utero* time-lapse movie with mNeonGreen::Tubulin and mCherry::H2B labeled.
(AVI)

**S7 Video. Metaphase through anaphase spindle morphology change in a strain with *zyg-8 (AID)* treated with auxin.** *In utero* time-lapse movie with GFP::Tubulin and mCherry::H2B labeled.
(AVI)

**S8 Video. Metaphase through anaphase spindle morphology change in a strain with *zyg-8 (or484)* at 25˚C.** *In utero* time-lapse movie with GFP::Tubulin and mCherry::H2B labeled.
(AVI)

**S9 Video. Anaphase A and B chromosome separation and spindle morphology change in a strain with *zyg-8(or484)* at 25˚C.** *In utero* time-lapse movie with GFP::Tubulin and mCherry::H2B labeled.
(AVI)

**S10 Video. Anaphase A and B chromosome separation and spindle morphology change in a strain with *zyg-8(or484)* and *spd-1(RNAi)* at 25˚C.** *In utero* time-lapse movie with GFP:: Tubulin and mCherry::H2B labeled.
(AVI)

## Acknowledgments

We thank Fede Pelisch, Sadie Wignall, Marie Delattre, Arshad Desai and the CGC, which is funded by NIH Office of Research Infrastructure Programs (P40 OD010440), for strains. We thank Yuxin Wang and Denisa Gabriela Lazureanu for assistance with strain generating. We thank Lesilee Rose and Richard McKenney for critical reading of the manuscript.

## Author Contributions

**Conceptualization:** Wenzhe Li, Francis J. McNally.

**Data curation:** Wenzhe Li.

**Formal analysis:** Wenzhe Li, Francis J. McNally.

**Funding acquisition:** Francis J. McNally.

**Investigation:** Wenzhe Li, Helena A. Crellin, Dhanya Cheerambathur.

**Project administration:** Francis J. McNally.

**Resources:** Helena A. Crellin, Dhanya Cheerambathur.

**Supervision:** Dhanya Cheerambathur, Francis J. McNally.

**Validation:** Wenzhe Li, Dhanya Cheerambathur.

**Writing – original draft:** Wenzhe Li, Francis J. McNally.

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
