## [Decision Letter · Decision Letter 0]

20 Jul 2023

Dear Dr McNally,

Thank you very much for submitting your Research Article entitled 'Redundant microtubule crosslinkers prevent meiotic spindle bending to ensure diploid offspring in C. elegans' to PLOS Genetics.

The manuscript was fully evaluated at the editorial level and by three independent peer reviewers. The reviewers appreciated the attention to an important problem, but raised some substantial concerns about the current manuscript. Based on the reviews, we will not be able to accept this version of the manuscript, but we would be willing to review a much-revised version. We cannot, of course, promise publication at that time.

Should you decide to revise the manuscript for further consideration here, your revisions should address the specific points made by each reviewer. In particular, we ask that you include: 1) clarifications of methods and conditions (i.e., whether data is from meiosis I, meiosis II, or both combined); 2) consider additional strategies to ZYG-8 knockdown given concerns about the use of GFP::ZYG-8 and *zyg-8(RNAi)* due to their partial loss-of-function and partial depletion, respectively; 3) address the issues with the use of *bmk-1(ok391)* instead of *bmk-1(syb3914) *raised by reviewers; and 4) include missing single mutants or depletions in graphs and panels. We will also require a detailed list of your responses to the review comments and a description of the changes you have made in the manuscript.

If you decide to revise the manuscript for further consideration at PLOS Genetics, please aim to resubmit within the next 60 days, unless it will take extra time to address the concerns of the reviewers, in which case we would appreciate an expected resubmission date by email to plosgenetics@plos.org.

We are sorry that we cannot be more positive about your manuscript at this stage. Please do not hesitate to contact us if you have any concerns or questions.

Yours sincerely,

Monica P. Colaiácovo

Academic Editor

PLOS Genetics

Gregory P. Copenhaver

Editor-in-Chief

PLOS Genetics

Reviewer's Responses to Questions

**Comments to the Authors:**

Reviewer #1: Li et al describe their analysis of multiple microtubule cross-linking proteins that stabilize the central spindle to drive anaphase B chromosome separation during C. elegans oocyte meiotic cell division. Using an extensive and rigorous genetic analysis with single, double and triple mutant oocytes, the authors provide evidence that while not being required individually, the conserved proteins KLP-19, SPD-1 and BMK-1 all participate in stabilization of the anaphase B spindle. The klp-19; bmk-1 and spd-1; bmk-1 double mutants both exhibited faster chromosome separation during anaphase B, indicating that both function redundantly with BMK-1 to limit the rate of separation. Curiously, klp-19; spd-1 double mutants exhibited slower separation and thus appear to operate in separate pathways that are each redundant with BMK-1 for limiting anaphase B separation rate. But oddly when the two pathways are eliminated in the presence of BMK-1 (in klp-19; spd-1 double mutants), the separation rate decreased, suggesting that KLP-19 and SPD-1 also can promote separation. Finally, the authors also show that another conserved microtubule binding protein called ZYG-8 promotes separation, with its loss leading to slower separation and also to decreased levels of both KLP-19 and SPD-1 at the spindle. Thus ZYG-8 appears to promote separation by competing with KLP-19 and SPD-1. The authors nicely discuss these results mechanistically in the context of the known biochemical properties of their conserved orthologs. These are very interesting results, and they provide a significant advance in our understanding of the proteins and mechanisms that stabilize the C. elegans oocyte meiotic spindle during chromosome separation. This stabilization is of particular interest given that cryoEM data have shown that these spindle are comprised of short overlapping microtubules that only rarely extend as single microtubules from the poles to the chromosomes, and hence it is evident that cross-linking and stabilization are needed. These results provide important new insight into the nature of this stabilization. However, before these results can be considered acceptable for publication in PLOS Genetics, the authors need to address the following concerns.

Major concerns.

1. The authors conclude that ZYG-8 promotes anaphase B chromosome separation by competing with both SPD-1 and KLP-19 for binding to spindle microtubules. However, the authors only see reduced anaphase B separation rates when using GFP RNAi on a CRISPR generated endogenously tagged GFP::ZYG-8 gene product, and not when using zyg-8 RNAi or zyg-8(or484ts). Furthermore, the authors only show reduced levels of SPD-1 and KLP-19 after RNAi knockdown of untagged endogenous ZYG-8.

As this is a key conclusion of the manuscript, the authors need to (1) demonstrate that an alternative approach to ZYG-8 knockdown also results in reduced anaphase B separation rates. Degron tagging of endogenous zyg-8 is easy to do with CRISPR (if one does not include GFP, and in this case it would be best not to include an FP tag but only the degron tag and assess knockdown effectiveness based on phenotype). Degron tagging generally provides strong knockdowns compared to RNAi, from observations in other studies thus far. The authors also could use zyg-8 RNAi in the zyg-8(or484ts) background at the restrictive temperature. The authors also should (2) show how much GFP RNAi and zyg-8 RNAi knock down the levels of GFP::ZYG-8, and (3) if zyg-8 RNAi reduces the already low levels of GFP::ZYG-8 to background, the authors should assess KLP-19 and SPD-1 levels after GFP RNAi in the GFP::ZYG-8 background with the other proteins also GFP tagged. Presumably one would see a greater reduction after GFP RNAi than by zyg-8 RNAi alone, given the difference in phenotypes seen with the two different approaches to reducing zyg-8 function. These are important experiments to do given the unlikely but real possibility that GFP tagging of ZYG-8 somehow alters its function in a neomorphic fashion, as the authors have only observed a requirement for anaphase using this GFP tagged allele.

2. Also with respect to ZYG-8, the authors state in lines 350-353 that reducing zyg-8 function (either with or484ts or via GFP RNAi in the GFP:ZYG-8 background) reduced anaphase A velocity, as documented in Fig S4G. In line 358, the authors also conclude that chromosome separation rate decreases after zyg-8 depletion, using the zyg-8 allele or484ts and specifying anaphase B in the text. However, Fig 5E shows increased velocity during anaphase A for zyg-8(or484) compared to control, and the difference appears significant (and certainly not decreased). The authors need to address these contradictory results (S4G vs 5E). Also, in S4G it appears that or484ts reduces anaphase A velocity as much or more than GFP RNAi, which is in contrast to the stronger effect with GFP RNAi for reducing anaphase B velocity, further highlighting the need to more conclusively examine ZYG-8 knockdown using a degron tag.

3. The authors combine data from both meiosis I and II, and yet there are substantial differences in oocyte meiotic spindle assembly during meiosis I and II, and moreover very little is known about meiosis II requirements given the earlier defects seen during meiosis I when analyzing gene functions. It is telling that the authors never even mention in the text that they are combining these data but mention it only in the Methods and never even refer to the supplemental figure showing the meiosis I and II comparisons except in the Methods! The authors should have been more transparent and should have indicated that they were combing meiosis I and II data within the Results section. Furthermore, in the supplemental figure showing the comparisons (S5), some of the conditions are quantified in only 2 or 3 or 4 oocytes, making the lack of statistical significance unreliable. It is confusing and creates a major caveat to combine data from meiosis I and II; the authors should provide only data for meiosis 1, or separate the data for meiosis I and II and provide sufficient examples to make the statistical analyses rigorous throughout.

4. The authors use the Student’s T test throughout the manuscript. Such a test is appropriate when the data show a normal distribution, which these kind of biological data usually do not show. The authors should either document normal distributions or preferably use the more rigorous U-test to better assess the significance of the differences they document in their comparisons.

5. In lines 267-269, the authors set out to test whether SPD-1 works in a complex with CYK-4 by depleting CYK-4 in the bmk-1(ok391) background. Because spd-1;bmk-1 double mutants show increased velocity, and the cyk-4;bmk-1 double mutant does not, they conclude the SPD-1 and CYK-4 do not act in a complex. This is reasonable logic, although the authors should state it more explicitly (they refer the lack of phenotype with cyk-4; bmk-1 and not how it differs from spd-1; bmk-1 double mutants). However, to be more conclusive, the authors should use the syb3914 allele of bmk-1, which they document is stronger than ok391. In general, it is not clear why the authors often use bmk-1(391) instead of syb3914, given that syb3914 showed increased anaphase B velocity while ok391 did not. In this case, to make the negative result more compelling, the authors should use the stronger bmk-1 allele syb3914. In addition, the authors should report the anaphase velocity when both CYK-4 and SPD-1 are reduced, as it is possible that a cyk-4; spd-1 double mutant might exhibit increased velocity if they function independently in this setting to stabilize the central spindle.

Minor comments.

1. In lines 163-164 the authors claim that BMK-1 does not localize to the poles during anaphase. However there does seem to be GFP:BMK-1 signal on the pole side of the chromosomes during anaphase A, both in Figure 1B and even more so in S1A where there is some yellow in the merge that is even centrally positioned.

2. The authors use two alleles of bmk-1, the partial deletion ok391 and the complete deletion syb3914, and the authors see an increase in anaphase B separate rate only with syb3914. However, in S2C the authors document a greater difference in anaphase B velocity when comparing ok391 with ok391;oj5, than they do when comparing syb3914 and syb3914; spd-1 RNAi. It would be helpful if the authors could document more consistency (in syb-3914 being stronger than ok391) by comparing syb3914 to syb3914;oj5, as they do with ok391.

3. The authors often leave out single mutants for their graphs showing velocity: S1E lacks klp-19 single mutant; Fig 2D only shows control and double mutant with neither single mutant; S2D does not show spd-1 RNAi or bmk-1(ok391) single mutants; Fig 3C lacks bmk-1(ok391) single mutant; 3E lacks bmk-1(ok391) and cyk-4 RNAi single mutants. It would be better to more consistently show all relevant single mutant genotypes.

4. In Figure 2, the authors distinguish between polar body not extruded, and polar body retracted. However, the live imaging frame examples shown for each polar body phenotype shows no difference between these two. Presumably one would have to show an early and late time point to document retraction. It is not clear that this difference matters (how long they did wait before deciding it didn’t retract; maybe some retracted later, etc.). It would be simpler to just call them all failed extrusion.

5. In Fig 5G, the authors quantify the time from the beginning of spindle rotation to the beginning of anaphase B. Does some spindle rotation occur during metaphase? Would it be better to simply quantify the time from the beginning of anaphase A (first time point when increased separation is detected) to beginning of anaphase (first time point when poles move apart)?

6. Some typos: line 181, which should read KLP-19:GFP; line 203 change Fig 1J-K to Fig 1K; line 271, change Fig S3C-D to Fig S3D; line 353 change S4F to S4G;

Reviewer #2: uploaded as attachment

Reviewer #3: See attachment

**Have all data underlying the figures and results presented in the manuscript been provided?**

Reviewer #1: Yes

Reviewer #2: None

Reviewer #3: None

PLOS authors have the option to publish the peer review history of their article (what does this mean?). If published, this will include your full peer review and any attached files.

Reviewer #1: No

Reviewer #2: **Yes: **Fede Pelisch

Reviewer #3: No

---

## [Decision Letter · Decision Letter 1]

5 Dec 2023

Dear Dr McNally,

We are pleased to inform you that your manuscript entitled "Redundant microtubule crosslinkers prevent meiotic spindle bending to ensure diploid offspring in C. elegans" has been editorially accepted for publication in PLOS Genetics. Congratulations!

As you prepare your final draft for the production team (the editorial team will not need to re-evaluate) we ask that you address the two minor comments made by Reviewer #3 (include the missing reference and update the methods section with the normalization strategy). You will also need to complete our formatting changes, which you will receive in a follow up email. Please be aware that it may take several days for you to receive this email; during this time no action is required by you. Please note: the accept date on your published article will reflect the date of this provisional acceptance, but your manuscript will not be scheduled for publication until the required changes have been made.

Yours sincerely,

Monica P. Colaiácovo

Academic Editor

PLOS Genetics

Gregory P. Copenhaver

Editor-in-Chief

PLOS Genetics

Comments from the reviewers (if applicable):

Reviewer's Responses to Questions

**Comments to the Authors:**

Reviewer #1: The authors have thoroughly addressed all reviewer comments and have added a substantial amount of new data and clarified the text in places. The manuscript is much improved and in my opinion is acceptable for publication in PLOS Genetics with no further revisions needed.

Reviewer #2: All the concerns/comments have been addressed adequately

Reviewer #3: The authors have done an excellent job responding to the feedback on the initial submission and the manuscript should be accepted for publication after addressing two minor issues:

1) The citation for the recent Dumont lab paper (PMID 37419936, Pitayu-Nugroho et al 2023) that analyzed klp-19(RNAi) is missing from the references section

2) The authors need to update their methods to add information on the normalization performed with the histone signal to correct for imaging depth.

**Have all data underlying the figures and results presented in the manuscript been provided?**

Reviewer #1: Yes

Reviewer #2: Yes

Reviewer #3: Yes

PLOS authors have the option to publish the peer review history of their article (what does this mean?). If published, this will include your full peer review and any attached files.

Reviewer #1: No

Reviewer #2: **Yes: **Fede Pelisch

Reviewer #3: No

**Data Deposition**

http://datadryad.org/submit?journalID=pgenetics&manu=PGENETICS-D-23-00663R1

**Press Queries**

---

## [Editor Report · Acceptance letter]

18 Dec 2023

PGENETICS-D-23-00663R1 

Redundant microtubule crosslinkers prevent meiotic spindle bending to ensure diploid offspring in *C. elegans*

Dear Dr McNally, 

We are pleased to inform you that your manuscript entitled "Redundant microtubule crosslinkers prevent meiotic spindle bending to ensure diploid offspring in *C. elegans*" has been formally accepted for publication in PLOS Genetics! Your manuscript is now with our production department and you will be notified of the publication date in due course.

With kind regards,

Lilla Horvath

PLOS Genetics

On behalf of:
